# A large accessory genome and high recombination rates may influence global distribution and broad host range of the fungal plant pathogen *Claviceps purpurea*

**Stephen Wyka[1], Stephen Mondo[1,2], Miao Liu[3], Vamsi Nalam[1], Kirk Broders[4,5]***

**1** Department of Agricultural Biology, Colorado State University, Fort Collins, Colorado, United States of America, **2** United States Department of Energy Joint Genome Institute, Berkeley, California, United States of America, **3** Ottawa Research and Development Centre, Agriculture and Agri-Food Canada, Ottawa, Canada, **4** USDA, Agricultural Research Service, National Center for Agricultural Utilization Research, Mycotoxin Prevention and Applied Microbiology Research Unit, Peoria, IL, United States of America, **5** Smithsonian Tropical Research Institute, Apartado Panamá, República de Panamá

* kirk.broders@usda.gov

**Data Availability Statement:** Most of the relevant data are within the paper and supporting files. Additional raw datasets and scripts are available on

## Abstract

Pangenome analyses are increasingly being utilized to study the evolution of eukaryotic organisms. While pangenomes can provide insight into polymorphic gene content, inferences about the ecological and adaptive potential of such organisms also need to be accompanied by additional supportive genomic analyses. In this study we constructed a pangenome of *Claviceps purpurea* from 24 genomes and examined the positive selection and recombination landscape of an economically important fungal organism for pharmacology and agricultural research. Together, these analyses revealed that *C. purpurea* has a relatively large accessory genome (~ 38%), high recombination rates (ρ = 0.044), and transposon mediated gene duplication. However, due to observations of relatively low transposable element (TE) content (8.8%) and a lack of variability in genome sizes, prolific TE expansion may be controlled by frequent recombination. We additionally identified that within the ergoline biosynthetic cluster the *lpsA1* and *lpsA2* were the result of a recombination event. However, the high recombination rates observed in *C. purpurea* may be influencing an overall trend of purifying selection across the genome. These results showcase the use of selection and recombination landscapes to identify mechanisms contributing to pangenome structure and primary factors influencing the evolution of an organism.

## Introduction

Pangenomes can provide useful insight into a species distribution and lifestyle through examination of gene functional diversity, abundance, and distribution into core and accessory genomes. These variations often provide fitness advantages and promote adaptive evolution of the organism [1–3]. In prokaryotes the existence of more open pangenomes (large accessory) has been suggested to be the result of adaptive evolution that allows organisms, with large

Dryad: Wyka, Stephen et al. (2020), A large accessory genome, high recombination rates, and selection of secondary metabolite genes help maintain global distribution and broad host range of the fungal plant pathogen Claviceps purpurea, v1, Dryad, Dataset, doi: https://doi.org/10.5061/dryad.6hdr7sqxp. Whole genome sequences generated for this project were deposited in NCBI BioProject PRJNA528707.

**Funding:** This work was supported by the Agriculture and Food Research Initiative (AFRI) National Institute of Food and Agriculture (NIFA) Fellowships Grant Program: Predoctoral Fellowships grant no. 2019-67011-29502/project accession no. 1019134 from the United States Department of Agriculture (USDA), the American Malting Barley Association grant no. 17037621, and the U.S. Department of Agriculture, Agricultural Research Service. Dr. Broders was supported in part by the Simon's Foundation Grant number 429440 to the Smithsonian Tropical Research Institute. The funders had no role in study design, data collection and analysis, decision to publish, or preparation of the manuscript.

effective population sizes, to migrate into new ecological niches [4]. Whereas closed pangenomes (larger core) are found to be associated with more obligate and specialized organisms [4]. Similar results have been identified in fungal species, where a range of saprotrophic to opportunistic yeasts were found to have accessory genomes representing ~ 9–19% of the genes [5], while *Zymoseptoria tritici*, a global wheat pathogen, has 40% of genes in the accessory genome [6]. This increase in the *Z. tritici* accessory genome reflects the global distribution of this pathogen that must continuously adapt to overcome new host resistances and multiple cycles of annual fungicide applications [6, 7]. While the identification of pangenome sizes provide valuable knowledge of polymorphic gene content, which can be used to infer the lifestyle of the species [4], a combination of pangenomic and alternative genomic analyses provide a deeper understanding of the primary factors that are contributing to pangenome structure and the adaptive trajectory of the organism.

*Claviceps purpurea* is a biotrophic ascomycete plant pathogen that has a specialized ovarian-specific non-systemic lifestyle with its grass hosts [8]. Despite the specialized infection pattern, *C. purpurea* has a broad host range of ~ 400 grass species across 8 grass tribes, including economically important cereal crops such as wheat, barley, and rye and has a global distribution [8]. However, the mechanisms that underlie the evolutionary success of this species is still understudied. Unlike other pathogens of cereal crops, researchers have been unsuccessful in identifying qualitative resistance genes in crop or wild grass varieties [9–11]. Menzies *et al.* [9] noted the potential for a complex virulence and host susceptibility relationship of *C. purpurea* on durum and hexaploid wheat varieties, however, virulence was determined if sclerotia weighed > 81 mg; indicating that *C. purpurea* is able to initiate its biotrophic interaction but might be arrested during the final stages of sclerotia development. During infection the fungus does not induce necrosis or hypersensitive response (host mediated cell death) in its host, instead it actively manages to maintain host cell viability to obtain nutrients from living tissue through a complex cross-talk of fungal cytokinin production [12–16]. Furthermore, Wyka *et al.* [17] revealed evidence of tandem gene duplication occurring in genes often associated with pathogenicity or evasion of host defenses (effectors), which may provide insight into the success of the species. However, the factors influencing these duplication events remain unclear.

*Claviceps purpurea* is also known for its diverse secondary metabolite profile of ergot alkaloids and pigments [18–21]. Fungal secondary metabolites can play important roles in plant-host interactions as virulence factors but can also increase the fitness of the fungus through stress tolerance [8, 22, 23]. It was recently postulated that the evolution of *C. purpurea* was associated with a host jump and subsequent adaptation and diversification to cooler, more open habitats [8, 17]. In addition, likely due to the toxicity of ergot alkaloids, grass grazing mammals showed avoidance in grazing grass infected with *C. purpurea*, suggesting a potential for beneficial effects for the host plant [24]. This along with other evidence of neutral to positive effects of infection to host plants [25, 26] suggest that *C. purpurea* is a conditional defensive mutualist [24].

In this study, we implement a comprehensive population genomic analysis to gain a deeper understanding of factors governing the evolution and adaptive potential of *C. purpurea*. Using 24 isolates, from six countries and three continents, we constructed the pangenome and subsequently identified genes with signatures of positive selection. Full genome alignments were further utilized to estimate population recombination rates and predict recombination hotspots. We observed a large accessory genome which may be influenced by a large effective population size and high recombination rates, which subsequently influence an overall trend of purifying selection and likely help defend against TE expansion. In addition, we observed that

**Table 1. Collection and annotation statistics for the 24 *Claviceps purpurea* genomes used in this study.**

| Strain ID[†] | Origin | Host | Genome size (Mb) | Genomic GC (%) | TE[‡] content (%) | Gene count | BUSCO[§] score (%) |
|---|---|---|---|---|---|---|---|
| LM46 | Canada: Alberta | *T. turgidum subsp. durum* | 30.6 | 51.80% | 9.64% | 8,455 | 97.00% |
| LM60 | Canada: Manitoba | *Avena sativa* | 30.6 | 51.70% | 9.29% | 8,498 | 97.10% |
| LM223 | Canada: Manitoba | *Bromus riparius* | 30.8 | 51.70% | 10.53% | 8,438 | 96.60% |
| LM207 | Canada: Manitoba | *Elymus repens* | 30.5 | 51.80% | 9.18% | 8,475 | 97.00% |
| LM5 | Canada: Manitoba | *Hordeum vulgare* | 30.5 | 51.80% | 8.95% | 8,508 | 97.40% |
| LM33 | Canada: Manitoba | *Hordeum vulgare* | 30.5 | 51.80% | 9.20% | 8,557 | 97.10% |
| LM232 | Canada: Manitoba | *Phalaris canariensis* | 30.7 | 51.70% | 9.36% | 8,512 | 96.70% |
| LM233 | Canada: Manitoba | *Phalaris canariensis* | 30.6 | 51.80% | 9.89% | 8,717 | 96.60% |
| LM4 | Canada: Manitoba | *Tricosecale* | 30.6 | 51.80% | 10.04% | 8,470 | 96.90% |
| LM470 | Canada: Ontario | *Elymus repens* | 30.5 | 51.80% | 8.95% | 8,591 | 96.80% |
| LM474 | Canada: Ontario | *Hordeum vulgare* | 30.6 | 51.80% | 9.38% | 8,500 | 97.20% |
| LM469 | Canada: Ontario | *Triticum aestivum* | 30.5 | 51.80% | 10.01% | 8,394 | 96.50% |
| LM461 | Canada: Quebec | *Elymus repens* | 30.5 | 51.80% | 8.42% | 8,656 | 97.30% |
| LM14 | Canada: Saskatchewan | *Hordeum vulgare* | 30.6 | 51.80% | 9.96% | 8,422 | 97.30% |
| LM30 | Canada: Saskatchewan | *Hordeum vulgare* | 30.6 | 51.80% | 9.35% | 8,526 | 96.30% |
| LM39 | Canada: Saskatchewan | *T. turgidum subsp. durum* | 30.5 | 51.80% | 10.11% | 8,591 | 97.00% |
| LM28 | Canada: Saskatchewan | *Triticum aestivum* | 30.6 | 51.70% | 9.58% | 8,713 | 97.00% |
| LM582 | Europe: Czech Republic | *Secale cereale* | 30.7 | 51.80% | 9.55% | 8,518 | 95.50% |
| 20.1 | Europe: Germany | *Secale cereale* | 32.1 | 51.60% | 10.87% | 8,703 | 95.50% |
| LM71 | Europe: United Kingdom | *Alopercurus myosuroides* | 30.5 | 51.80% | 9.59% | 8,472 | 97.00% |
| Clav55 | Oceania: New Zealand | *Lolium perenne* | 30.7 | 51.80% | 9.80% | 8,480 | 97.00% |
| Clav04 | USA: Colorado | *Bromus inermis* | 31.8 | 51.70% | 10.05% | 8,824 | 97.70% |
| Clav26 | USA: Colorado | *Hordeum vulgare* | 30.8 | 51.70% | 9.07% | 8,737 | 98.00% |
| Clav46 | USA: Wyoming | *Secale cereale* | 30.8 | 51.70% | 9.68% | 8,597 | 97.10% |

† NCBI BioProject: PRJNA528707 (except 20.1, NCBI Accession = SAMEA2272775).

‡ Transposable element content presented in [17], as a proportion of genomic sequences.

§ Benchmarking Universal Single-Copy Orthologs Dikarya database (odb9).

the *lpsA1* and *lpsA2* genes of the well-known ergoline biosynthetic cluster were likely the result of a recombination event.

## Materials and methods

### Genome data

Haploid genome data from a collection of 24 isolates was utilized in this study to provide a comprehensive analysis of *Claviceps purpurea*. The 32.1 Mb reference genome of *C. purpurea* strain 20.1 was sequenced in 2013 using a combination of single and paired-end pyrosequencing (3 kb fragments) resulting in a final assembly of 191 scaffolds [18; NCBI: SAMEA2272775]. The remaining 23 isolates were recently Illumina sequenced, assembled, and annotated in [17, 27; NCBI BioProject: PRJNA528707], representing a collection of isolates from USA, Canada, Europe, and New Zealand (Table 1). To define gene models for our subsequent analyses, the reference genome was subject to an amino acid cutoff of 50 to match the other 23 isolates. In this study, we report the pangenome of *C. purpurea*, analysis of the population genomic recombination, and the landscape of genes with signatures of positive selection.

Gene functional and transposable element (TE) annotations utilized were those reported in Wyka *et al.* [17] and datasets Wyka *et al.* [27]. In brief, secondary metabolite clusters were predicted using antiSMASH v5 [28], with all genes belonging to identified clusters classified as secondary (2˚) metabolites. Functional domain annotations were conducted using InterProScan v5 [29], HMMer v3.2.1 [30] search against the Pfam-A v32.0 and dbCAN v8.0 CAZYmes databases, and a BLASTp 2.9.0+ search against the MEROPs protease database v12.0 [31]. Proteins were classified as secreted proteins if they had signal peptides detected by both Phobius v1.01 [32] and SignalP v4.1 [33] and did not possess a transmembrane domain as predicted by Phobius and TMHMM v2.0 [34]. Effector proteins were identified by using EffectorP v2.0 [35] on the set of secreted proteins for each genome. Transmembrane proteins were identified if both Phobius and TMHMM detected transmembrane domains. Transposable elements fragments were identified following procedures for establishment of *de novo* comprehensive repeat libraries set forth in Berriman *et al.* [36] through a combined use of RepeatModeler v1.0.8 [37], TransposonPSI [38], LTR_finder v1.07 [39], LTR_harvest v1.5.10 [40], LTR_digest v1.5.10 [41], Usearch v11.0.667 [42], and RepeatClassifier v1.0.8 [37] with the addition of all curated fungal TEs from RepBase v24.03 [43]. RepeatMasker v4.0.7 [37] was then used to identify TE regions and soft mask the genomes. These steps were automated through construction of a custom script, TransposableELMT (https://github.com/PlantDr430/TransposableELMT) [17, 20].

### Pangenome analysis

The pangenome was constructed using OrthoFinder v2.3.3 [44], on all genes identified from the 24 genomes, to infer groups of orthologous gene clusters (orthogroups). OrthoFinder was run using BLASTp on default settings. For downstream analysis, gene clusters were classified as secreted, predicted effectors, transmembrane, secondary (2˚) metabolites, carbohydrate-degrading enzymes (CAZys), proteases (MEROPs), and conserved domain (conserved) clusters if $\geq$ 50% of the strains present in a gene cluster had at least one protein classified as such. Gene clusters not grouped into any of the above categories were categorized as unclassified.

Core and pangenome size curves were extrapolated from resampling of 24 random possible combinations for each pangenome size of 1–24 genomes and modelled by fitting the power law regression formula: $y = Ax^B + C$ using the curve_fit function in the Python module Scipy v1.4.1. These processes were automated through the creation of a custom python script (https://github.com/PlantDr430/FunFinder_Pangenome).

### Positive selection

To investigate the positive selection landscape of genes we utilized the 53 isolates (22 species) of the *Claviceps* genus [17, 20; NCBI BioProject: PRJNA528707] and found single-copy orthologs using OrthoFinder v2.3.3 with BLASTp on default settings. A total of 3,628 single-copy orthologs were identified (See Table 2 for detailed report). For each ortholog cluster sequences were aligned using MUSCLE v3.8.1551 [45] on default settings and values of dN, dS, and dN/dS (omega, ω) were estimated using the YN00 [46] method in PAML v4.8 using default parameters.

For statistical purposes, each gene cluster was only characterized by one functional category in the order displayed in Table 2 (i.e. secreted genes are those not already classified as effectors, etc) (See Methods section *Statistical analyses and plotting*).

### Genome alignment, SNP calling, and recombination

Procedures followed [47], for creation of a fine-scale recombination map of fungal organisms and identification of recombination hotspots. A brief description will be provided below, for a more detailed methodology and explanation of algorithms refer to [47–49].

**Table 2. PAML processing information and filtering of core orthogroups for calculation of dN/dS (ω) ratios.**

| Total gene clusters (Pangenome) | 10,540 | | | |
|---|---|---|---|---|
| Single-copy gene clusters (Pangenome) | 6,244 | | | |
| Single-copy gene clusters (Claviceps genus) | 3,628 | | | |
| Number of clusters with N/A PAML results | 33 | | | |
| Cluster Classification (non-redundant) †: | Total Pangenome | Total Core‡ | Single-copy genes (Pangenome)‡ | Single-copy genes (*Claviceps* genus)‡ |
| Effectors | 257 | 100 (38.9%) | 84 (32.7%) | 13 (5.1%) |
| Secreted | 366 | 278 (75.9%) | 253 (69.1%) | 109 (29.8%) |
| 2˚ Metabolites | 313 | 202 (64.5%) | 181 (57.8%) | 78 (24.9%) |
| Transmembrane | 1,210 | 998 (82.5%) | 949 (78.4%) | 567 (46.9%) |
| MEROPs | 167 | 149 (89.2%) | 143 (85.6%) | 89 (53.3%) |
| CAZys | 75 | 68 (90.7%) | 66 (88.0%) | 36 (48.0%) |
| Conserved | 4,754 | 3,985 (83.8%) | 3,808 (80.1%) | 2,390 (50.3%) |
| Unclassified | 3,398 | 778 (22.9%) | 717 (21.1%) | 320 (9.4%) |

† For statistical purposes classification is structured such that each cluster is only represented once (in the order provided), i.e. secreted clusters are those not already classified as effectors, etc.

‡ Percentage out of total pangenome.

LastZ and MultiZ from the TBA package [50] was used to create the population genome alignment projected against the reference genome, *C. purpurea* strain 20.1 [18]. Alignments in MAF format were filtered using MafFilter v.1.3.1 [51] following [47]. Final alignments were merged according to the reference genome and subsequently divided into nonoverlapping windows of 100 kb. MafFilter was additionally used to compute genome-wide estimates of nucleotide diversity (Watterson's θ) and Tajima's D in 10 kb windows. Single nucleotide polymorphisms (SNPs) were called by MafFilter from the final alignment. Principal Component Analysis (PCA) and a Maximum-Likelihood phylogeny were conducted with fully resolved biallelic SNPs (Table 3) using the R package SNPRelate v1.18.1 [52] and RAxML v8.2.12 [53] using GTRGAMA and 1000 bootstrap replicates, respectively.

The following process was automated through the creation of a custom python script (https://github.com/PlantDr430/CSU_scripts/blob/master/Fungal_recombination.py). LDhat [54] was used to estimate population recombination rates (ρ) from the filtered alignment using only fully resolved biallelic positions. A likelihood table was created for the θ value 0.01, corresponding to the genome-wide Watterson's θ of *C. purpurea* (Table 3; Julien Dutheil *per comm*), and LDhat was run with 10,000,000 iterations, sampled every 5000 iterations, with a burn-in of 100,000. The parameter ρ relates to the actual recombination rate in haploid organism through the equation $\rho = 2N_e \times r$, *where $N_e$ is the effective population size and r is the per site rate of recombination*. However, without knowledge of $N_e$ we cannot confidently infer *r* and thus sought to avoid the bias of incorrect assumptions. Therefore, we reported the population recombination rate (ρ).

Resulting recombination maps were filtered to remove pairs of SNPs for which the confidence interval of the recombination estimate was higher than two times the mean [47]. Average recombination rates were calculated in regions by weighing the average recombination estimate between every pair of SNPs by the physical distance between the SNPs. Using the reference annotation file [18], we calculated the average recombination rates for features in each gene: 1) exons, 2) introns, 3) 500 bp upstream, and 4) 500 bp downstream with a minimum of three filtered SNPs. Flanking upstream and downstream regions correspond to the 5´ and 3´ regions for forward stranded genes and the 3´ and 5´ regions for reverse stranded genes. We

**Table 3. Summary statistics of whole-genome alignment filtering and SNP calls for *Claviceps purpurea*.**

| *C. purpurea* strain 20.1 | | |
|---|---|---|
| Number of scaffolds | 191 | |
| Size of reference genome (bp) | 32,091,443 | |
| Number of exonic sites in reference genome (bp) | 12,774,951 (39.8%) | |
| Number of haplotypes | 24 | |
| Summary Genome alignment: | Total Alignment Length (bp) | Number of alignment blocks |
| MultiZ alignment | 27,523,755 | 16,330 |
| Keep blocks with all strains | 27,517,978 | 15,861 |
| MAFFT in 10kb windows | 27,378,024 | 15,870 |
| Filter 1 | 26,198,304 | 57,891 |
| Filter 2 | 24,959,120 | 97,532 |
| Merged per contigs (N's filled in) | 31,389,412 | 154 |
| Total number of SNPs | 1,152,999 | |
| Total number of analyzed SNPs (biallelic, no unresolved state) and percent of total SNPs | 1,076,901 (93.4%) | |
| Total number of SNPs in exons and percent of total | 370,045 (32.1%) | |
| Total number of analyzed SNPs in exons (biallelic, no unresolved state) and percent of total analyzed SNPs in exons | 358,258 (96.8%) | |
| Diversity in 10kb windows: | Median | |
| Watterson's Θ | 0.01196 | |
| Tajima's D | -0.82522 | |

also calculated the average recombination rate for each intergenic region between the upstream and downstream regions of each gene. Introns were added to the GFF3 file using the GenomeTools package [55]. The original recombination maps produced from LDhat (Julien Dutheil *per comm*) were converted from bp to kb format for use in LDhot [48] to detect recombination hotspots with 1000 simulations and the parameter—windlist 10 was used to create 20 kb background windows [49]. Only hotspots with a value of ρ between 5 and 100 and width < 20 kb were selected for further analysis [47–49].

## Statistical and enrichment analyses

Statistics and figures were generated using Python3 modules SciPy v1.3.1, statsmodel v0.11.0, Matplotlib v3.1.1, and seaborn v0.10.0. All multi-test corrections were performed with Benjamini-Hochberg false discovery rate procedure. Enrichment analyses were tested using Fischer's Exact test with a cutoff α = 0.05. Uncorrected p-values were corrected using Benjamini-Hochberg and Bonferroni multi-test correction with a false discovery rate (FDR) cutoff of α = 0.05. Corresponding p-values from correction tests were averaged together to get a final p-value. Enrichment was performed on protein domain names and GO terms. Orthogroups were only associated with a domain or GO term if ≥ 50% of the strains present in the gene cluster had one gene with the term. This process was automated through creation of a custom python script (https://github.com/PlantDr430/CSU_scripts/blob/master/Domain_enrichment.py).

## Results

### Pangenome analysis

We constructed a pangenome of *Claviceps purpurea* from 24 isolates representing a collection from three continents and six countries (Table 1). Taking advantage of plentiful isolates

available from Canada, we sampled more heavily from different provinces and on different host plants. The principal component and phylogenetic analysis revealed substantial genetic variation among the samples. However, the genetic distances were not correlated with geographic distances, such as LM470 (Canada) and Clav04 (USA) grouping closer to isolates from Europe and the isolate from New Zealand (S1 Fig). In addition, across Canada and USA, isolates from similar regions rarely clustered together and were often intermixed (S1B Fig). These results agree with the results from a multi-locus genotyping of a larger set of samples from Canada and USA [56]. Previous reports [17] showed that *C. purpurea* isolates had similar genome size (30.5 Mb– 32.1 Mb), genomic GC content (51.6% - 51.8%), TE content (8.42% - 10.87%), gene content (8,394–8,824), and BUSCO completeness score (95.5% - 98.0%) (Table 1). The pangenome consisted of 205,354 genes which were assigned to 10,540 orthogroups. We observed 6,558 (62.22%) orthogroups shared between all 24 isolates (core genome), of which 6,244 (59.2%) were single-copy gene clusters, while the remaining core orthogroups, 314 (3%), contained paralogs (2–8 paralogs per cluster). The accessory genome consisted of 3,982 (37.78%) orthogroups with 2,851 (27.05%) shared by at least two isolates (but not all) and 1,131 (10.73%) were lineage-specific (singletons) found in only one isolate (Fig 1 and S1 Table). Within the accessory genome (including lineage-specific orthogroups) we observed 592 (5.6%) orthogroups containing paralogs, with some isolates containing > 20 genes per cluster (Fig 1C and S1 Table).

We utilized multiple gene functional categories to get a deeper understanding of how genes of different function were structured within the pangenome. As a proportion of orthogroups within each pangenome category (core, accessory, and singleton) we found that the core genome was significantly enriched in orthogroups that contained genes with conserved protein domains (conserved) (5,471; 84%), transmembrane domains (transmembrane) (1,038; 16%), peptidase and protease domains (MEROPs) (211, 3.2%), and orthogroups of carbohydrate-active enzymes (CAZys) (212, 3.2%) ($P < 0.01$, Fisher's exact test, Fig 2A and 2E–2G). Effector proteins play major roles in plant-microbe interactions, often conveying infection potential of the pathogen. A total of 257 predicted effector orthogroups were identified; 100 (38.9%) were core, 143 (55.6%) were accessory, and 14 (5.4%) were singletons. Predicted effectors and orthogroups coding for secreted proteins, which also contribute to host-pathogen interactions, were significantly enriched in the accessory genome (143, 5%; 218, 7.6%; respectively) ($P < 0.01$, Fisher's exact test, Fig 2C and 2D). Although, the accessory and singleton genomes were largely composed of unclassified orthogroups (1791, 62.8%; 830, 73.4%; respectively) ($P < 0.01$, Fisher's exact test, Fig 2H). Lastly, we observed that orthogroups which contained secondary (2°) metabolite genes were similarly represented across all pangenome categories ($P > 0.05$, Fisher's exact test, Fig 2B).

As expected, core orthogroups were found to be significantly enriched in general housekeeping and basic cellular functions and development such as protein and ATP binding, nucleus and membrane cellular components, and transmembrane transport, metabolic, and oxidation-reduction processes (S2 Table). Protein domains in core orthogroups were significantly enriched for several WD40-repeat domains, P-loop nucleoside triphosphate hydrolase (IPR027417), armadillo-type fold (IPR016024), and a major facilitator (PF07690) (S2 Table). When narrowing the focus to orthogroups with paralogs, core paralogous orthogroups were enriched in cytochrome P450 domains, and domains associated with trehalose activity (S3 Table). In contrast, the accessory genome was only found to be enriched in a fungal acid metalloendopeptidase domain (MER0001399) and the singleton genome had enrichment for a Tc5 transposase DNA-binding domain (PF03221) (S2 Table). Accessory paralogs were found to be enriched in several protein kinases, Myb-like domains, phosphotransferases, as well as DNA integration and a MULE transposase domain (S3 Table). It should be noted that the high

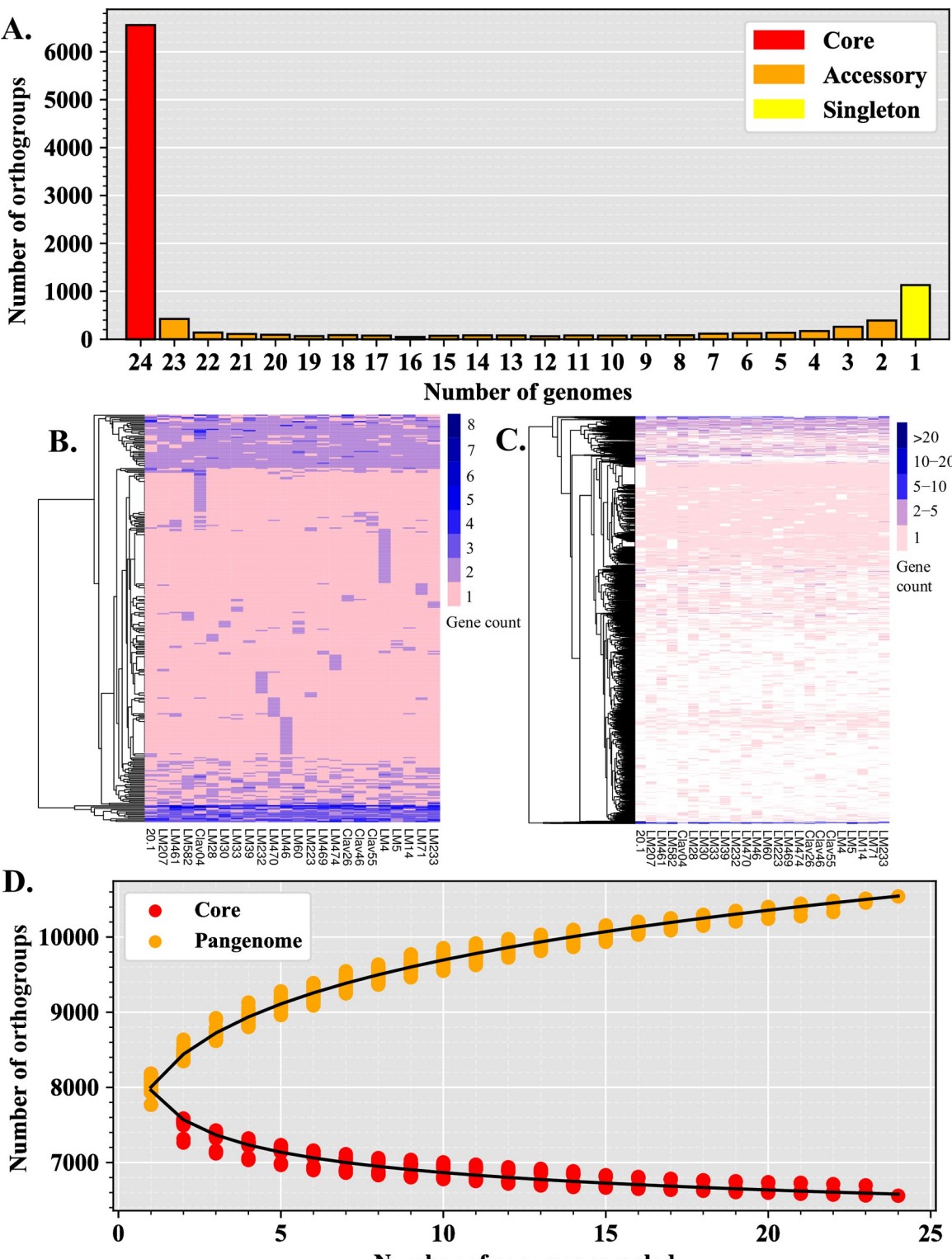

**Fig 1. The pangenome of *Claviceps purpurea*. (A)** Categorization of orthogroups (gene clusters) into core (shared between all isolates), accessory (shared between ≥ 2 isolates, but not all), and singletons (found in only one isolate) according to the number of orthogroups shared between genomes. **(B)** Copy number variation in core orthogroups containing paralogs. **(C)** Presence/absence variation and copy number variation of accessory orthogroups, not including singletons. **(D)** Estimation of core and pangenome (core + accessory + singleton) sizes by random resampling of possible combinations of 1–24 genomes (dots). Curves were modelled by fitting the power law regression formula: y = Ax$^B$ + C.

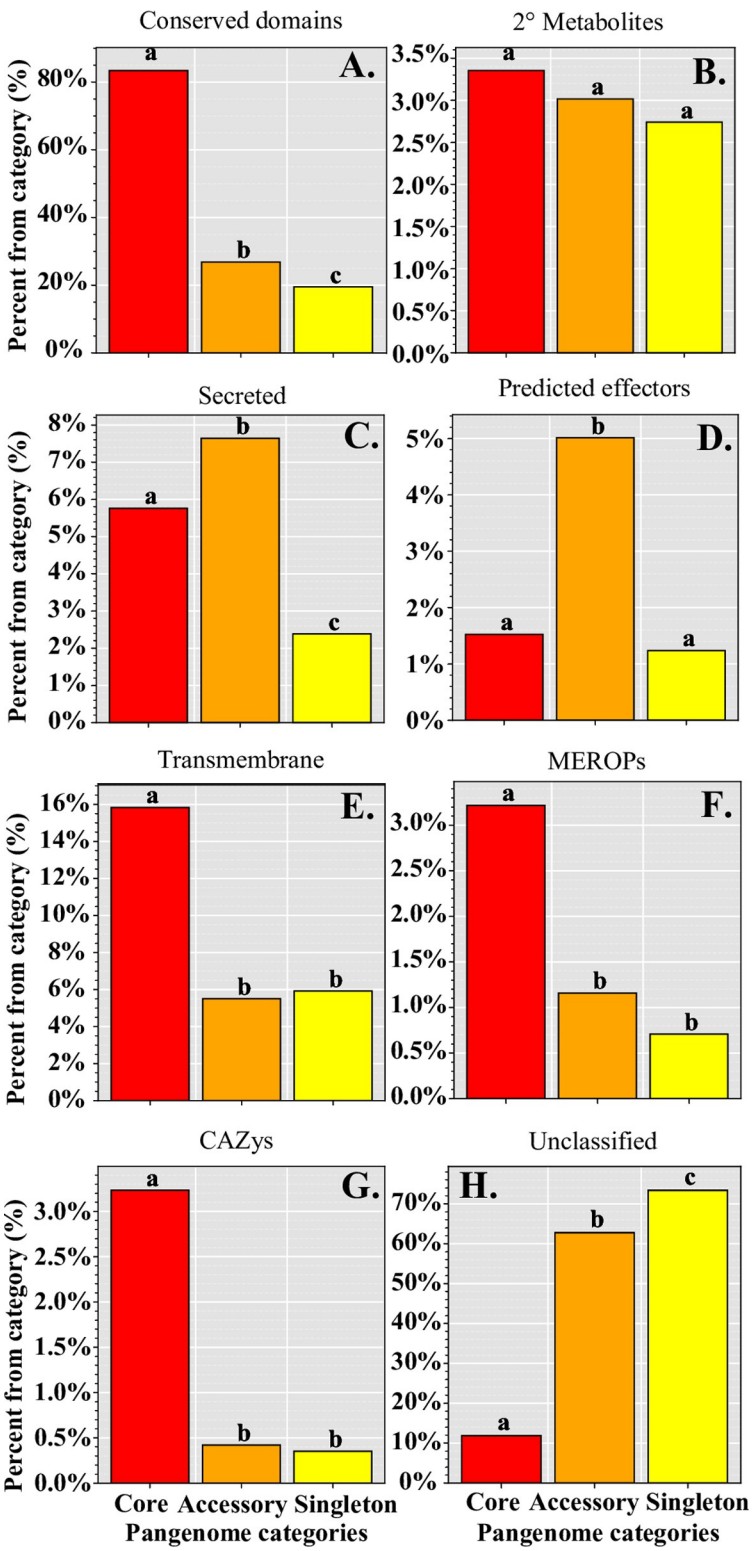

**Fig 2. Analysis of predicted protein function across the *Claviceps purpurea* pangenome.** Graphs indicate the proportion of orthogroups within each pangenome category of classified protein function. (**A**) Containing conserved protein domains, (**B**) genes found in secondary (2˚) metabolite clusters, (**C**) possessing predicted secreted signals, (**D**) predicted to be effectors, (**E**) containing transmembrane domains, (**F**) containing MEROPs domains for proteases and peptidases, (**G**) contain CAZY enzymes, (**H**) all unclassified orthogroups not falling into a previous category. Different

letters (within each classification) represent significant differences determined by multi-test corrected Fisher exact test ($P < 0.01$).

abundance of unclassified genes in the accessory genome may have increased the level of type II error rates for GO and domain enrichment analyses. Overall, our results revealed a large accessory pangenome enriched with genes associated with host-pathogen interactions (predicted effectors) and an abundance of orthogroups containing paralogs (8.6%), indicating the presence of proliferate gene duplication occurring within the species.

## Selection landscape

To further understand the evolution of genes within the pangenome we investigated the positive selection landscape on protein coding genes using 3,628 single-copy core orthologs to compute the ratio of non-synonymous substitutions to synonymous substitutions (dN/dS) (Table 2). Ratios of dN/dS (omega, ω) can provide information of evolutionary forces shaping an organism as genes with $\omega > 1$ may indicate positive or diversifying selection, $\omega = 1$ may indicate neutral evolution, and $\omega < 1$ may indicate negative or purifying selection [57].

Overall, we saw low dN ($0.047 \pm 0.046$) and high dS ($0.37 \pm 0.16$) values across all functional categories (S3 Fig), corresponding to low ω ratios (Fig 3). This suggests a general trend of purifying selection within *C. purpurea*, with only 8 (0.2%) orthogroups with ω values $> 1$, of which 6 (0.16%) were functionally unclassified (Fig 3 and S4 Table). Of the two genes with $\omega > 1$ and functional annotations, one was a high compatibility group (HMG) box domain containing protein (OG0003348, $\omega = 1.17$) and the other contained a Type IIB DNA topoisomerase domain and was related to meiotic recombination protein *rec12* (OG0003965, $\omega = 1.52$) (S5 Table). Overall, core unclassified genes showed significantly higher ω values than all other functional categories ($P < 0.05$, multi-test corrected Mann-Whitney U Test, Fig 3). In contrast, transmembrane, MEROPs, CAZys, and secondary (2˚) metabolites showed significantly lower ω values ($P < 0.05$, multi-test corrected Mann-Whitney U Test, Fig 3) compared to the other

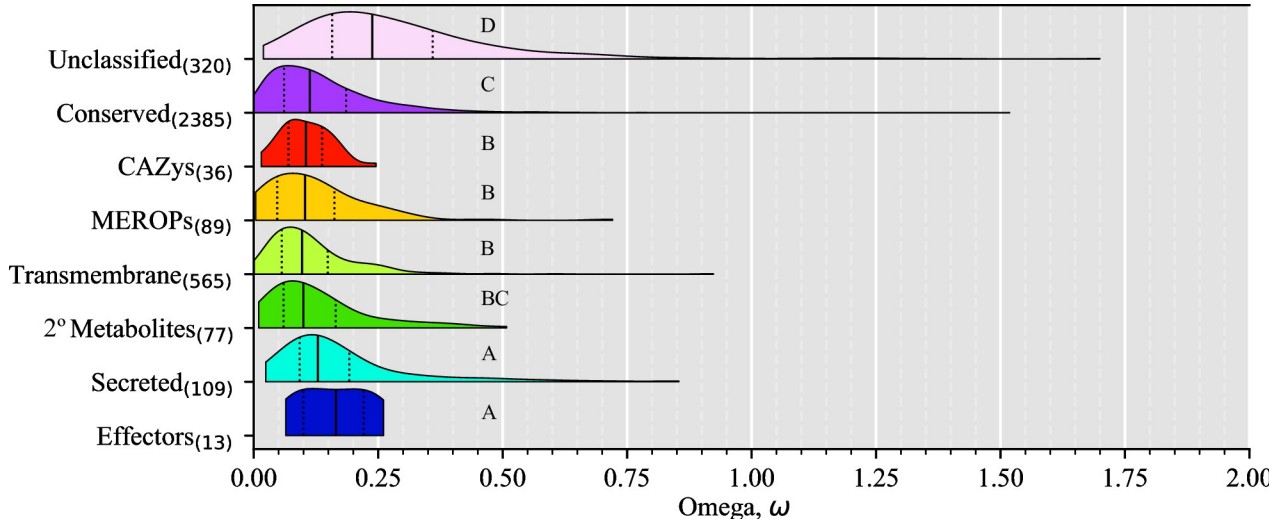

**Fig 3. Distribution of omega (ω, dN/dS) ratios within the *Claviceps purpurea* core genome.** Violin plots of ω ratios for core single-copy orthogroups protein functional categories. Solid vertical lines within each plot represent the median, while dotted lines represent the 25th and 75th quartile, respectively. Different letters represent significant differences determined by Kruskal-Wallis with *post hoc* multi-test corrected Mann-Whitney U Test ($\alpha \leq 0.05$).

functional categories (except 2˚ metabolites which was not significantly different than conserved genes), indicating that these genes are frequently experiencing purifying selection.

## Recombination landscape

Recombination is also an important potential driver of genome evolution and plays a central role in the adaptability of parasitic organisms to overcome host defenses [58]. Our genome-alignments contained 154 of the original 191 scaffolds (Table 3). The 37 missing scaffolds totaled 222,918 bp (average lengths = 6,192 ± 5,676 bp) and corresponded to 59 genes. Thirty-one of the missing scaffolds contain genes that were only part of the accessory genome of which six scaffolds contained two or more genes (S6 Table), suggesting that these scaffolds represent blocks of genetic material that could be lost or gained from isolate to isolate. Only 1 of the missing scaffolds did not contained any genes. Most of the genes found on these scaffolds encoded conserved domains associated with either reverse transcriptase, integrases, or helicases (S6 Table), which suggest unplaced repetitive content. Although, one scaffold (scaffold 185) did possess a gene encoding a conserved domain for a centromere binding protein (S6 Table). Together these observations may indicate the potential for dispensable chromosomes, as dispensable and mini-chromosomes often contain higher repetitive content [59], however, long-read sequencing is necessary for confirmation.

From our shared alignments of all 24 genomes, we recovered 1,076,901 biallelic SNPs corresponding to a median nucleotide diversity (Watterson's θ) of 0.01196 and a Tajima's D of -0.82522 calculated from 10 kb non-overlapping windows (Table 3). The resulting SNPs were used to infer the population recombination rate (ρ) from the linkage disequilibrium between SNPs based on *a priori* specified population mutation rate θ, which was set to 0.01 based on our nucleotide diversity (Watterson's θ) (Table 3) [47]. The *C. purpurea* genome recombination landscape was highly variable as some scaffolds showed highly heterogenous landscapes, other scaffolds showed intermixed large peaks of recombination, while others still had more constantly sized peaks across the regions (Fig 4 and S4 Fig). Overall, the mean genomic population recombination rate in *C. purpurea* was ρ = 0.044. Recombination in specific sequence features and gene type were examined through comparison of mean population recombination rates in exons, introns, 500-bp upstream and downstream of the coding DNA sequence, and intergenic regions based on the annotation of the reference genome (strain 20.1). The distribution of population recombination rates was comparable across different gene features and gene functional categories, although, some significant differences were observed (Fig 5). In general, we found upstream regions to have the lowest recombination rates, while downstream regions have the highest recombination rates (Fig 5). The decreased recombination in upstream regions might be the result of mechanisms trying to conserve promotor regions. This trend was observed across different functional gene categories, except in predicted effector genes where exons showed the highest recombination rates and downstream regions with the lowest, although these were not significantly different (Fig 5B). Across functional categories, secreted genes and transmembrane genes showed the highest recombination rates within each gene feature but were not always significantly different (Fig 5C).

Due to the observation of paralogs (Fig 1) and evidence of tandem gene duplication in *C. purpurea* [17] the extent recombination might have influenced these events was investigated. Duplicated genes were found to have lower population recombination rates than all other genes within the genome (Fig 5D), suggesting that other factors are influencing gene duplication. Due to the absence of repeat-induced point (RIP) mutation [17], transposable elements (TEs) are likely a contributing factor. To investigate the association of duplicated genes with TEs we calculated the average distance of genes to transposons (DNA and long terminal repeat

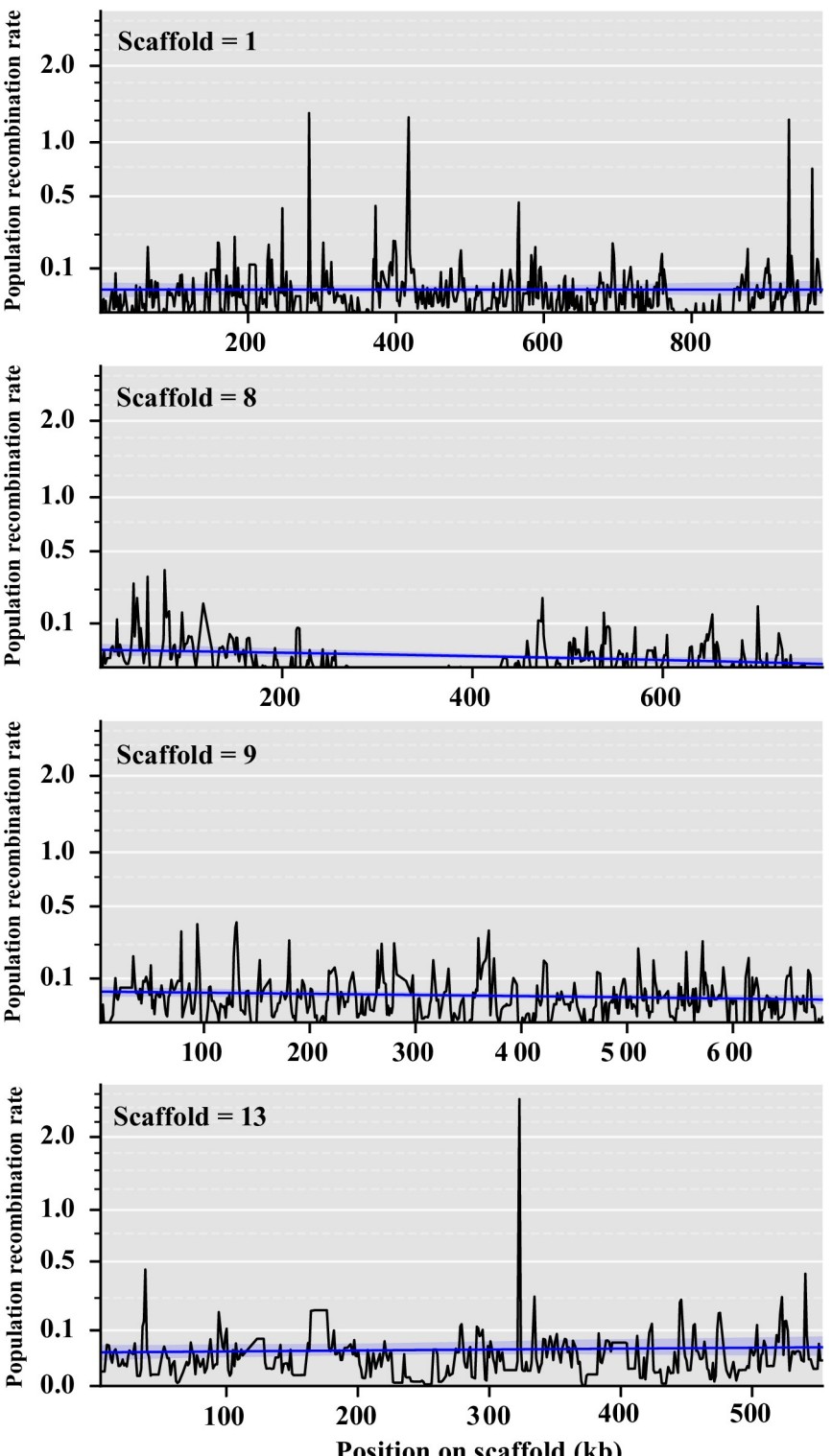

**Fig 4. Population recombination rates of representative scaffolds.** Estimates of population recombination rates ($\rho$), in non-overlapping 1 kb windows, across four representative scaffolds displaying the different variation observed across the *Claviceps purpurea* genome. Smoothing curves were calculated from population recombination rates in 10 kb windows. See S4 Fig for remaining scaffolds.

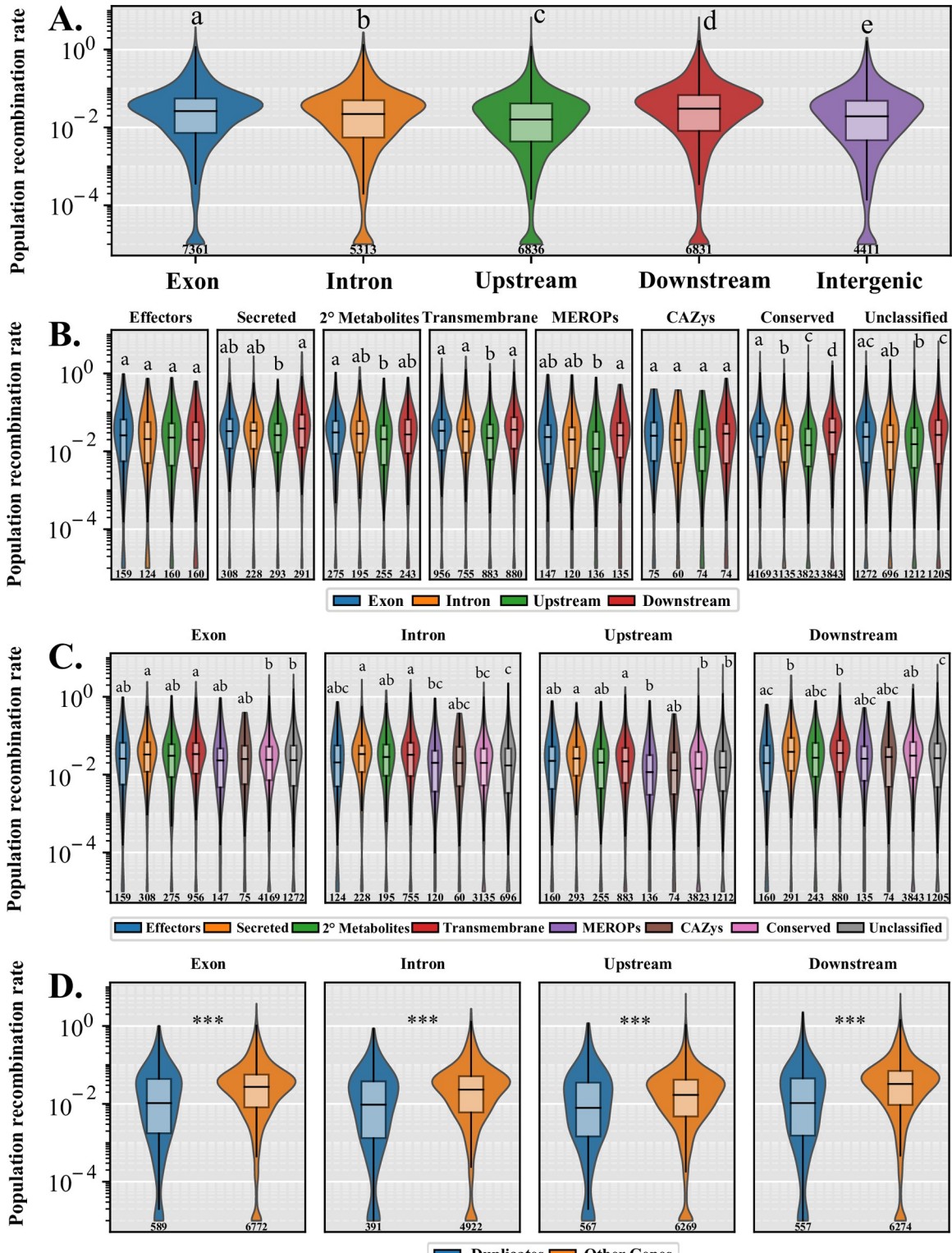

**Fig 5. Fine-scale recombination patterns across the *Claviceps purpurea* genome.** Plots indicate the distribution of estimated population recombination rates (ρ) between **(A)** different gene features (exons, introns, 500bp upstream and downstream), and **(B-D)** genes of different functional categories and classification. Different letters represent significant differences determined by Kruskal-Wallis with *post hoc* multi-test corrected Mann-Whitney U Test (α ≤ 0.01) between data within each plotting window, *** $P < 0.0001$. Sample sizes are embedded below each plot.

(LTR) retrotransposons) and the average number of flanking transposons. Results showed duplicated genes were significantly closer to LTRs and had significantly more flanking LTRs than predicted effectors and other genes ($P < 0.0001$, multi-test corrected Mann-Whitney U Test, S5 Fig). In addition, for all genes examined LTRs were significantly closer than DNA transposons (genes ($P < 0.0001$, multi-test corrected Mann-Whitney U Test, S5 Fig).

As distinct peaks of recombination were observed (Fig 4 and S4 Fig), LDhot was used to call statistically significant recombination hotspots by analysis of the intensity of recombination rates in 3 kb (1 kb increments) windows compared to background recombination rates in 20 kb windows [47–49]. After implementing a cutoff of $\rho \geq 5$ and length of 20 kb [48] only five recombination hotspots were retained, ranging from 11 kb to 18.5 kb in length (Fig 6). A recombination hotspot was identified between the *lpsA1* and *lpsA2* genes of the ergoline biosynthetic cluster, suggesting that this gene duplication event was likely the result of recombination (Fig 6D). Association of gene functional category and TEs within hotspots varied between regions. Some hotspots showed a greater association with duplicated genes and TEs (Fig 6B–6D), while others showed no association with duplicated genes (Fig 6E) or no association with TEs (Fig 6A). In general, genes with conserved protein domains showed the highest presence within hotspots (S6 Fig). It should be noted that some unclassified genes and genes with conserved protein domains associated with hotspots were also found to be overlapping regions identified as repeats (Fig 6A–6C and 6E). Protein domains found within these genes were associated with ankyrin (IPR002110) and tetratricopeptide (IPR013026) repeats. Only 5 of the 846 duplicated genes [reported in 17] found throughout the reference genome were located within predicted recombination hotspots (Fig 6 and S6 Fig). While Wyka et al. [17] showed that gene cluster expansion was prevalent among predicted effectors, only one non-duplicated predicted effector (CCE30212.1) was found located within a recombination hotspot (Fig 6C). Together these results suggest that while recombination may result in important gene duplication, it is not the primary driver of gene duplication within *C. purpurea*.

## Discussion

The establishment of a *Claviceps purpurea* pangenome from 24 isolates, as well as the detection of core genes with signatures of positive selection and analysis of the recombination landscape have provided knowledge into how high recombination rates and gene duplication are driving the genomic evolution and adaptation of the species.

The pangenome of *C. purpurea* reveals a large accessory genome with 37.78% accessory orthogroups (27.05% accessory + 10.73% singleton) in comparison to four model fungal pangenomes (*Saccharomyces cerevisiae*, *Candida albicans*, *Cryptococcus neoformans*, and *Aspergillus fumigatus*), which found around 9–19% of their genes in the accessory genome [5]. Our results are more comparable to the pangenome of the fungal pathogen *Zymoseptoria tritici* which had an accessory genome comprised of 40% (30% accessory + 10% singleton) of genes [6]. Similar to *C. purpurea*, *Zymoseptoria tritici* is a globally distributed fungal pathogen of wheat, suggesting that fungal species with similar geographical distributions could possess comparable pangenome structures as they are under similar evolutionary pressures. Shared ecological habitats and lifestyles have been reported to influence pangenome sizes in bacteria [4]. In fact, *C. purpurea* and *Z. tritici* both experienced enrichment of predicted effector orthogroups in the accessory genome and enrichment of carbohydrate-active enzymes (CAZys) orthogroups in the core genome (Fig 2) [6], conveying a comparable similarity between gene functions as both organisms are pathogens of wheat. In addition, Badet et al. [6] suggested that the large accessory genome of *Z. tritici* is likely maintained due to TE activity and a large effective population size as a result of observations of high SNP density, rapid

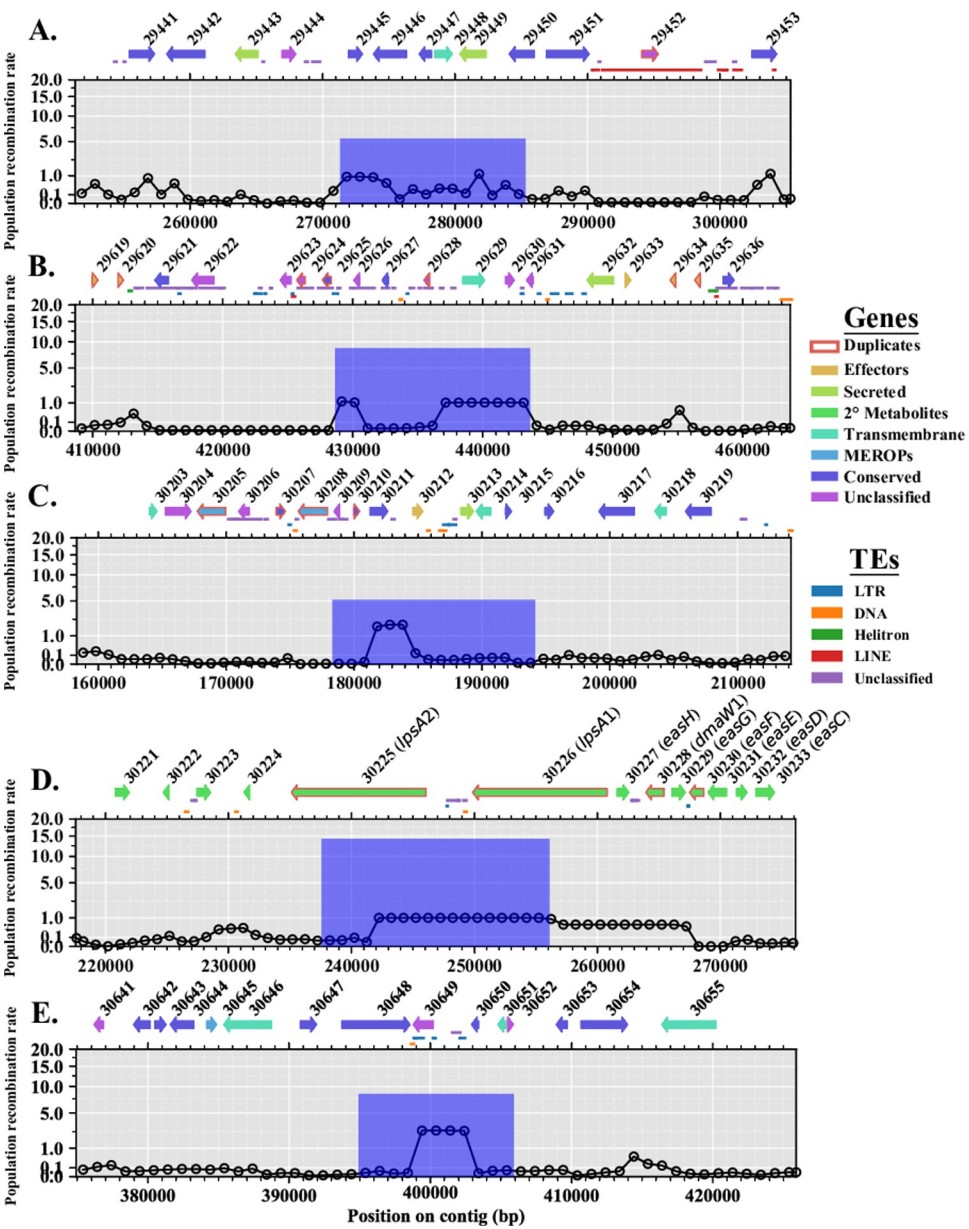

**Fig 6. Recombination hotspots predicted in *Claviceps purpurea* with associated genes and transposable elements (TEs).** Panels indicate scaffolds: **(A)** scaffold 14; **(B)** scaffold 15; **(C, D)** scaffold 20; **(E)** scaffold 23. Lines indicate background population recombination rates (ρ) estimated in non-overlapping 1 kb windows. Blue bars represent the position, intensity, and width of the predicted hotspots. Genes within the hotspot window and surrounding (± 20 kb) region are depicted by arrows with protein ID's of the reference (strain 20.1) from NCBI. Genes identified as duplicated (≥ 80% identity) from Wyka *et al.* 2021 are outlined in red. TEs are depicted by lines between genes and the corresponding hotspot graph. Colors of arrows and lines correspond to the legend on the right.

decay in linkage disequilibrium, and high recombination rates [47, 60, 61]. The same mechanisms could also explain the large accessory genome observed in *C. purpurea*.

We identified 37 missing scaffolds in our population genome alignment with 31 of these containing genes only present in the accessory genome, suggesting the potential for blocks of DNA that could be lost/gained between isolates. Of these accessory scaffolds 15 contained genes encoding conserved domains associated with either reverse transcriptase, integrases, or

helicases and one scaffold possessed a gene encoding a conserved domain for a centromere binding protein (S6 Table). Together these could indicate the potential for dispensable mini-chromosomes, as dispensable and mini-chromosomes often contain higher repetitive content [59]. However, these unplaced contigs may be assembly artifacts. Due to our Illumina based assemblies we did not process these elements further but believe that these are important aspects of *C. purpurea* evolution and should be a focal point of future research with the advantage of long-read sequencing to understand their function more confidently. Due to these transcriptase rich unplaced scaffolds, the lack of RIP, and observation of TEs with 0% divergence [17], we believe transposons and/or transcriptases are influencing the evolution of the accessory genome in *C. purpurea*.

We observed an abundance of orthogroups containing paralogs (8.6%). This presence of gene duplication and association with LTR retrotransposons (S5 Fig) could be contributing to the large size of the accessory genome, potentially through pseudogenization and/or neofunctionalization. In fact, unclassified genes had the highest ω (dN/dS) ratios (Fig 3). In addition, the abundance of duplication in accessory unclassified genes [17] and their small sizes (S2 Fig) further suggests the presence of pseudogenization and/or neofunctionalization. Badet *et al.* [6] suggested that TEs were likely contributing to the *Z. tritici* accessory genome due to the correlation of TE content with genome size and observations of transcribed TEs. We observed a similar correlation of TE content with genome size ($P = 0.004$, Adj. $R^2 = 0.28$), however, our genome sizes and TE content (30.5 Mb– 32.1 Mb, 8.42% - 10.87%, respectively) were not as variable as in *Z. tritici*, which also had a twofold higher TE content [6]. This suggests that TEs play a more important role in *Z. tritici* genome expansion, however, only 0.2% of the orthogroups in *Z. tritici* contained paralogs suggesting that gene duplication is not as common in *Z. tritici* as it is in *C. purpurea* (8.6% paralogs). The lack of gene duplication in *Z. tritici* is likely due to the presence of RIP [62], which should also reduce TE expansion through silencing [63–65]. While we lack RNAseq data to observe TE transcription within *C. purpurea*, observations of TEs with 0% divergence in *C. purpurea* [17] suggest recent TE activity. The observed reduced association of recombination with duplicated genes (Fig 6D) and association of duplicated genes with LTR transposons (S5 Fig) would suggest that gene duplication in *C. purpurea* is mediated in part by transposon activity.

Due to the potential for transposon mediated gene duplication, it was remarkable to find relatively low TE content (~8–10%) within *C. purpurea*, especially in the absence of RIP. Other genomic mechanisms, such as recombination, may limit TE expansion and increases in genome size. Tiley and Burleigh [66] found a strong negative correlation between global recombination rate, genome size and LTR retrotransposon proportion across 29 plant species, indicating that higher recombination rates actively reduce genome size likely through the removal of LTR elements. A similar function may be affecting LTR content in *C. purpurea*, which would explain the observed differences in LTR content between *Claviceps* section *Claviceps* (low LTR content, RIP absent) and *Claviceps* sections *Pusillae*, *Paspalorum*, and *Citrinae* (high LTR content, RIP present) [17].

On average we observed a twofold higher mean population recombination rate (ρ = 0.044) in *C. purpurea* than *Z. tritici* (ρ = 0.0217) and tenfold higher than *Z. ardabiliae* (ρ = 0.0045) [47]. As ρ is a function of effective population size and recombination rate per site (ρ = $2N_e \times r$), these increases could be the result of the increment in recombination rate per site (*r*) and/or effective population size ($N_e$). Differences in ρ between the two *Zymoseptoria* species was postulated to be due to increased recombination rates per site as it was found that the nucleotide diversity (Watterson's θ = $2 N_e$ x μ, where μ is mutation rate) was 1.6 times higher in *Z. tritici* (0.0139) than *Z. ardabiliae* (0.00866). Under an assumption that both *Z. tritici* and *Z. ardabiliae* have comparable mutation rates, $N_e$ of *Z. tritici* would only be 1.6 times higher than *Z.*

*ardabiliae*, therefore, the 5-fold higher ρ would likely be caused by higher recombination rates per site [47]. Our observed Watterson's θ of 0.012 in *C. purpurea* (Table 2) is comparable to *Z. tritici*, suggesting that if mutation rates and effective populations sizes are comparable than the twofold increase in ρ is likely influenced by higher recombination rates per site in *C. purpurea*. Although, *Z. tritici* is a heterothallic organism while *C. purpurea* is homothallic [67] but *C. purpurea* does frequently out-cross in nature [19, 68], suggesting that these factors may provide a difference in effective population sizes between these organisms. In addition, mutation rates might be higher in *Z. tritici*, than *C. purpurea*, due to the presence of RIP, which identifies repeat/duplicated sequences within a genome and introduces C:G to T:A mutations to effectively silence these regions [63–65]. It has also been reported that RIP can "leak" into neighboring non-repetitive regions and introduce mutations, thus, accelerating the rate of mutations, particularly those in closer proximity to repeat regions [69–71]. If the mutation rate is increased in *Z. tritici* due to RIP "leakage" the nucleotide diversity in *Z. tritici* could be the result of high mutation rates, whereas the nucleotide diversity in *C. purpurea* could be influenced by higher effective population size and/or recombination rates per site. Our positive selection analysis did reveal a gene with evidence of positive selection (ω = 1.52) that is related to meiotic recombination protein *rec12*, which is known to catalyze the formation of dsDNA breaks that initiate homologous recombination in meiosis in yeast [72]. Kan *et al.* [72] further determined that the frequency of dsDNA breaks catalyzed by *rec12* significantly increased the frequency of intergenic recombination. In both plants [66] and *Z. tritici* [73], higher recombination rates were found to increase the efficacy of purifying selection. Similarly, *C. purpurea* had an overall trend of purifying selection with skewness towards lower ω values (Fig 3) and an observed correlation of higher population recombination rates around genes with lower ω ratios (S7 Fig), further suggesting the potential for higher recombination rates in *C. purpurea*.

Additional support for higher recombination rates per site in *C. purpurea* could be extrapolated from recombination hotspots, or lack thereof. While we observed evidence of a heterogenous recombination landscape with several scaffolds showing large peaks in population recombination rates (Fig 4 and S4 Fig), we only predicted five recombination hotspots (Fig 6), which is in stark contrast to the ~1,200 hotspots identified in *Z. tritici* [74]. On average, we did observe higher population recombination rates across scaffolds compared to the rates observed across chromosomes of *Zymoseptoria* [47], suggesting that the background recombination rate in *C. purpurea* is higher and "flatter", potentially limiting the detection of hotspots [48]. Overall, this indicates that *C. purpurea* exhibits high recombination rates per site, which potentially helps defend against TE expansion.

While these higher recombination rates are likely influencing the purifying selection observed in regions of *C. purpurea* genome, it may not be the only reason we were unable to detect predicted core effector genes with signatures of positive selection (Fig 3). Specifically, it is possible that positive selection is occurring but the whole gene dN/dS ratio is <1. Predicted effectors may have sites under positive selection, but that are not detected by this whole gene analysis of positive selection. Therefore, our results represent a lack of power to positive selection in predicted effector genes, and not actual evidence for a lack of selection. Another potential explanation is that the ancestral state of *Claviceps purpurea* is plant endophytism [8] and is closely related to several mutualistic grass endophytes (i.e. *Epichloe*, *Balansia*, *Atkinsonella*) which have been known to provide beneficial aspects to their hosts mostly through production of secondary metabolites and plant hormones [75–77]. Furthermore, Wäli et al. [24] classified *C. purpurea* as a conditional defense mutualist with its plant host, as they found sheep avoided grazing infected grasses and observed that infection rates were higher in grazed pastures compared to ungrazed fields. Other researchers have observed neutral to positive effects of seed set, seed weight, and plant growth on infected plants compared to uninfected plants [24–26, 78].

These factors, along with the broad host range of *C. purpurea* (400+ grass species) and lack of known crop resistance (R) genes, could suggest a lack of strong selection for resistance to *C. purpurea* in grass species [24]. This could help explain the lack of positive selection observed in predicted core effector genes, implying that effectors are not under strong selection pressure to compete in the evolutionary arms race against host defense. However, it should be noted that positive selection analyses are computed from single-copy core orthologs. Observations of significant enrichment of predicted effector genes in the accessory genome of *C. purpurea* and duplication of effector gene cluster [17] could implicate their role in diversity of infection potential [7], however, no host specific races of *C. purpurea* have been identified.

While further research is needed to better characterize the accessory genome of *C. purpurea* it appears that TE mediated-gene duplication and frequent recombination are likely playing a role in the expansion of *C. purpurea's* accessory genome and may be influencing the success of *C. purpurea*. In addition, all members of *Claviceps* section *Claviceps*, which contain grass pathogens that have extended geographical distributions and host ranges, have genomes that lack RIP, exhibit gene duplication, and have comparable TE content [17], suggesting that the genomic mechanisms identified in this study might be characteristic of section *Claviceps* as a whole.

## Conclusion

Overall, we observed that ~38% of the *Claviceps purpurea* pangenome is accessory, which is likely influenced by a large effective population size, frequent recombination, and TE mediated gene duplication. Pseudogenization and neofunctionalization might also be contributing due to the observed TE activity, observations of higher ω ratios, signatures of positive selection in core single-copy unclassified genes, and small size of many accessory unclassified genes. Due to a lack of RIP, prolific TE expansion is likely kept under controlled by high recombination rates, which subsequently may be influencing the overall trend of purifying selection.

## Supporting information

**S1 Fig. Genetic diversity of 24 *Claviceps purpurea* isolates.**
(TIF)

**S2 Fig. Average protein lengths (aa) of all orthogroups in *Claviceps purpurea* pangenome.**
(TIF)

**S3 Fig. Distributions of mean non-synonymous (dN) and synonymous (dS) substitution rates of core single-copy orthogroups in *Claviceps purpurea*.**
(TIF)

**S4 Fig. Estimated population recombination rates of *Claviceps purpurea* scaffolds.**
(TIF)

**S5 Fig. Distributions of genes and their association (distance and flanking counts) to LTR transposable elements.**
(TIF)

**S6 Fig. Association of genes within recombination hotspots.**
(TIF)

**S7 Fig. Correlation of recombination rates and omega ratios.**
(TIF)

**S1 Table. *Claviceps purpurea* pangenome spreadsheet.**
(XLSX)

**S2 Table. Enrichment of protein domains within pangenome.**
(XLSX)

**S3 Table. Enrichment of protein domains within paralogous orthogroups.**
(XLSX)

**S4 Table. PAML summarized results.**
(XLSX)

**S5 Table. BLAST results of single-copy core orthologs with an ω (dN/dS) ≥ 1.**
(XLSX)

**S6 Table. Annotation information of missing reference scaffolds from 24 isolate whole-genome alignment.**
(XLSX)

## Acknowledgments

We would like to thank Julien Dutheil for his assistance in understanding the procedures for estimating fungal recombination rates using LDhat and LDhot.

## Author Contributions

**Conceptualization:** Stephen Wyka, Miao Liu, Kirk Broders.

**Data curation:** Stephen Wyka.

**Formal analysis:** Stephen Wyka, Stephen Mondo.

**Funding acquisition:** Kirk Broders.

**Methodology:** Stephen Wyka, Stephen Mondo.

**Project administration:** Vamsi Nalam, Kirk Broders.

**Resources:** Miao Liu.

**Supervision:** Vamsi Nalam, Kirk Broders.

**Writing – original draft:** Stephen Wyka.

**Writing – review & editing:** Stephen Wyka, Stephen Mondo, Miao Liu, Vamsi Nalam, Kirk Broders.

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
