## [Decision Letter · Decision Letter 0]

19 Mar 2021

PONE-D-21-02186

A large accessory genome, high recombination rates, and selection of secondary metabolite genes help maintain global distribution and broad host range of the fungal plant pathogen Claviceps purpurea

PLOS ONE

Dear Dr. Broders,

Thank you for submitting your manuscript to PLOS ONE. After careful consideration, we feel that it has merit but does not fully meet PLOS ONE’s publication criteria as it currently stands. Therefore, we invite you to submit a revised version of the manuscript that addresses the points raised during the review process.

You will see I took the unusual step of assigning myself as a reviewer of this manuscript. I am fully disclosing this, so as not to violate any PLOS ONE policies. I had read through your manuscript completely before inviting reviewers, but it was not until receiving 2 independent reviews and going over the manuscript again before I realized I had missed something very important in your positive selection analysis methods that the 2 reviewers apparently missed as well. Unless the methods have not been explained fully, the use of the dN/dS analyses on the intra-species sample will not give valid estimates for selection pressures, affecting a large portion of the manuscript. Because the manuscript would require a major rewrite and potentially lots of reanalysis, I'm not sure whether Major Revisions or Open Reject is a better decision for the manuscript, though I have selected Major Revision. Suffice it to say, there are aspects of the manuscript and data that you present that will be useful to other researchers, and I encourage the manuscript to be revised/resubmitted. If the timeline for resubmission due to the Major Revisions decision is not possible, the rewritten manuscript can be submitted as a new manuscript, and I would be happy to again serve as academic editor.

Besides this major problem with the manuscript, the other reviewers agreed that the data presented in the paper at most partly supported your conclusions. I agree with another reviewer in that your discussion includes many speculations without robust results supporting the claims, so care should be taken in the wording of such passages to reflect what is possible rather than you believe your data clearly show [required]. Another specific point to address [required] is Reviewer #1's concern about your accessory chromosome evidence (Reviewer #2 also had similar concerns regarding resolving indels within chromosomes from accessory mini-chromosomes without longer sequencing reads). One aspect of this is the question of assembly quality, and thus what evidence scaffolds absent in some isolates actually represent chromosomes. The other is related to the global nature of the sample, such that some presence/absence differences might reflect divergence between separate populations - though it is not clear such cases of presence/absence between populations would not be classified as accessory genome sequences. You should also provide some general clarification on the statistical testing and power of the study, though I don't necessarily see a major problem here. I don't see any need to remove citations to tables and figures from the Discussion section - this stylistic choice is up to you, even if it is not all that common.

We look forward to receiving your revised manuscript.

Kind regards,

Christopher Toomajian

Academic Editor

PLOS ONE

Journal Requirements:

3. Please include your tables as part of your main manuscript and remove the individual files. Please note that supplementary tables should remain as separate "supporting information" files.

Reviewers' comments:

Reviewer's Responses to Questions

**Comments to the Author**

1. Is the manuscript technically sound, and do the data support the conclusions?

Reviewer #1: Partly

Reviewer #2: Partly

Reviewer #3: No

2. Has the statistical analysis been performed appropriately and rigorously? 

Reviewer #1: I Don't Know

Reviewer #2: Yes

Reviewer #3: Yes

3. Have the authors made all data underlying the findings in their manuscript fully available?

Reviewer #1: Yes

Reviewer #2: Yes

Reviewer #3: Yes

4. Is the manuscript presented in an intelligible fashion and written in standard English?

Reviewer #1: Yes

Reviewer #2: Yes

Reviewer #3: Yes

5. Review Comments to the Author

Reviewer #1: Major concerns:

252-254: I would be more supportive about the evidence for accessory chromosome(s) if these data were derived from the population-level accessory pangenome. In this study the accessory pangenome references all isolates gathered across the globe – global populations may not exchange genetic material, so 1) How can you distinguish between accessory chromosome scaffolds and genomic regions that evolved from isolation? 2) How can you distinguish them from poorly assembled scaffolds? Short reads are especially prone to not assembling well, potentially leaving streaks of N’s in scaffolds derived from small contigs... particularly from regions with low coverage. At this point, the study points to accessory chromosomes - my assessment is that accessory chromosomes are not substantiated by these data and the other 2 options should be presented as the assumed nulls.

Other concerns:

639-646: I’m concerned about the type 1 error thresholds – considering the individual type 1 error thresholds, I think the cumulative type 1 error for the study could be large. The data seems justified visually and corrections are in place in explicit multiple comparisons analyses, but the statistical power of the overall study concerns me. https://www.ncbi.nlm.nih.gov/pmc/articles/PMC4840791/

Use of normative language without reference points, e.g. vague adjectives as in line 45 “broad genomic analyses”; general grammatic errors/tense switching/active-passive voice switching, e.g. 89 “the factors that were influencing”

Abstract:

“which is often governed by variable gene content.” - what is governed by variable gene content, what does this mean, what information does this add?

Data locations:

“Dataset, doi: XXX” – remove or add doi

Author Summary:

43-44: “ambiguity surrounding the true nature” is vague, please be specific

Introduction:

*88-89: how does tandem gene duplication implicate their positive contribution to Claviceps – how is this definitively not just a random duplication? Looking for a citation here or explanation of how

106: “use SCOs to identify genes under positive selection” – the SCOs don’t show which genes are under positive selection, what was the actual method that references the SCOs?

Results:

183-184: here, the percentage in ‘()’ is relative to the whole then the next is relative to the previous percentage; previous orthogroup percentages in the text had a common reference point. I think readers could infer, so I can see arguments either way, but I would argue either maintain common reference points or explicitly state them so the reader doesn’t have to infer.

185: “two conserved genes” – do you mean orthogroups?

224: again, I’m not sure I’d say “genes” when you are referencing what appear to be orthogroups… If there are paralogs within these orthogroups then I don’t think it is appropriate to universally asign a function to the OG because paralogs can take on different functions. At present, it might be appropriate if they are single copy orthologs, but I don’t think that is made clear if that’s the case.

252-254: see major concerns

*255-270: are these obtained from individual populations or the entire sample set?

282: RIP is introduced without expanding the acronym.

Discussion:

*338-339: how is it supported that this is potentially due to a lack of RIP?

417-420: this is not substantiated; perhaps instead of saying “it is plausible to believe”, state the two hypotheses you have mentioned. At present, this is added without any support.

Conclusions:

503-504: This section would benefit from reducing normative language i.e. “large” is used twice, “high” is used - there’s no reference to base these off. Use actual values and if you want to compare with other fungi then actually some sort of reference

Methods:

639-646: see general concerns

Reviewer #2: PONE-D-21-02186

A large accessory genome, high recombination rates, and selection of secondary metabolite genes help maintain global distribution and broad host range of the fungal plant pathogen Claviceps purpurea

The authors conducted a comprehensive study on the extended gene repertoire (pangenome) of Claviceps purpurea sampled from different hosts in six countries. The inclusion of samples from different geographical locations and the total number of 24 genomes would be adequate for exploring for the first time the pangenome of this pathogen. Overall, the results presented bring novelty and would be fit for publication in PLOS One after major revisions. Major criticisms are regarding long parts of the discussion being mostly speculations without robust results for supporting their claims. Therefore, the discussion needs to focus on the major novelties presented (i.e., positive selection in secondary metabolites) and shortened in other sections (i.e., mini-chromosome speculation, several comparisons with Zymoseptoria tritici). In my detailed comments below, I request some clarifications and suggest modifications in the manuscript.

The manuscript figures are of high quality, and the authors provide open access to scripts and genomic data, which should be praised.

Specific comments organized according to line numbers in the manuscript:

Results

FigS1: What is the impact of the hosts of origin on the population structure?

FigS1: Mention the enlarged view within the PCA.

Line 126: Liu et al not referenced. Would it be possible to provide preprint?

Line 131: Is 59.2% correct?

Line 151: Please clarify what would be the consequences of unclassified orthologs in the accessory genome for the GO enrichment.

Line 200: I understand the details for the “stringent filtering” are in the methods, but briefly mention here.

Line 220: Given that effectors in the core genome are not under selection in the present study, how authors see the role of effectors in the accessory genome?

Line 252: Since authors used Illumina reads mapped on a not complete (chromosome level) reference genome, how to distinguish whether different scaffolds would ultimately compose a single chromosome and represent indels instead of mini-chromosomes? Also, what would be the impact of structural variations in the pangenome (core + accessory + indels)?

Line 283: Why have authors not considered DNA transposons?

Line 298: “Some hotspots showed a greater association with duplicated genes and TEs (Fig. 7 B-D), while others showed a lower association (Fig. 7 A, E).” ….. If I’m interpreting the figure correctly, I’d suggest being more explicit for Figure 7A = no TE’s within the recombination hotspot.

Discussion

As a general comment, please remove references to tables and figures.

Line 363 to 382 is mainly speculative as long reads would be necessary to tackle the mini-chromosome hypothesis. Would it be possible to remove/shorten this section?

Line 324: Zymoseptoria tritici is not a biotrophic fungus (see references: doi.org/10.1016/j.fgb.2015.04.001and
doi.org/10.1094/phi-i-2011-0407-01), and the only grass it infects is wheat (Please see http://www.genome.org/cgi/doi/10.1101/gr.118851.110 and also Seifbarghi et al 2009). Please, authors should correct this statement and discussion based on this wrong definition. Please clarify that the global distribution of Zymoseptoria tritici is due to wheat infection and that grasses (other than wheat) are infected by different Zymoseptoria.

Line 325: Please tone down comparisons with Zymoseptoria tritici. Z. tritici is heterothallic, has a very limited host range, and develops across a crop season mostly via asexual spores (and hyphae).

Line 343: Is there RNAseq data available for improving gene annotation? Considering the signatures of positive selection on unclassified genes, how would positive selection act on pseudogenes in C. purpurea?

Line 428: Populations in areas without fungicide pressure or deployment of resistant cultivars (Israel, Oregon) have similarly high levels of genetic diversity when compared to regions with intensive usage and deployment (Switzerland, Oregon R) as for example in the study of Hartmann et al. 2017. I suggest authors to shorten this discussion section to avoid speculations.

Line 435: Are C. purpurea of different hosts able to cross sexually? How would it influence population recombination rates and hotspots analyses?

Line 446: Are high levels of recombination expected in a homothallic fungus?

Line 478: I see this section on the evolution of secondary metabolites in C. purpurea as a novel and relevant discussion that should receive focus by shorting previous sections.

L 488: Reference about Puccinia missing.

Materials and Methods

Table 1: I could not find LM28 and 582 in the PRJNA528707.

Line 522: Mention the sequencing technology for the 23 isolates.

Line 524: “cutoff of 50 aa” What does it mean?

Line 546: What RepBase version?

Line 567: Describe how gene sequences of individual genomes were extracted for selection analyses.

Reviewer #3: The authors use resequenced, assembled, and annotated genomes from 24 C. purpurea isolates to perform a pangenome analysis, and combine this work with two additional analyses, estimating recombination rates (and detecting hotspots), and performing an analysis of selection on coding sequences (single copy orthologous genes). There are many different approaches to infer the action of natural selection in population genetics datasets, and they authors here estimate selection (purifying and positive selection) through estimating dN/dS (omega) for their population set.

The work presented here involves a fairly large dataset, as well as lots of bioinformatics analysis, and so those contributions alone can serve as important resources for understudied species. That said, these large-scale genomic analyses are not that well integrated in this manuscript (at least they are integrated to some extent, especially through various enrichment analyses), and other population genetic analyses could have been included that might help to integrate the different pieces in other ways. The interpretation of population genomic studies is not straightforward, and results tend to be consistent with multiple explanations until more specific analyses are carried out. In this work, the results tend to be over-interpreted, and in many cases the evidence for the authors' conclusions are rather weak. However, unless some important methods descriptions have been left out of this manuscript, the biggest problem in the manuscript is the analysis of natural selection. It appears the authors have carried out the dN/dS analyses with PAML and the CodeML algorithm using only the 24 C. purpurea isolates and without inclusion of data from additional species. The Jeffares et al. paper they cite indicates sequences from multiple species are needed for these analyses [states that PAML FAQ recommends a minimum 4-5 species, refers to multiple alignments of protein-coding gene sequences from several species in a phylogeny, in describing required input files indicates an annotated genome for at least 1 related species is needed besides the target species of interest]. dN/dS analyses are meant to compare nonsynonymous versus synonymous substitutions (alleles gone to fixation in some species or independent lineages), yet what we have in a populations capable of sexual reproduction are segregating polymorphisms (since the isolates are not independent lineages), and the distinction is important. Kryazhimskiy and Plotkin (https://www.ncbi.nlm.nih.gov/pmc/articles/PMC2596312/) have shown that inferring selection pressures from dN/dS calculated from population samples does not work in the same way that dN/dS analyses work when multiple species are compared (the behavior of dN/dS is expected to be different). In particular, they find that even for genes experiencing negative selection, the observation of an elevated dN/dS (values closer to 1) is expected with intra-specific samples. And the observation of dN/dS<1 is also consistent with strong positive selection. Given this problem, the major report from the manuscript that secondary metabolism genes are more important than effectors based on the dN/dS analyses must be called into question, since it no longer has any reliable support.

The selection analyses represent a major section of the manuscript - lines 174-238 in the Results, 566-588 in the Methods, Figures 3 and 4, Table 2, Supplemental figures S3, S5, S9, and supplemental tables S4-S7 all are based on the highly questionable dN/dS results. To become publishable, the manuscript should either remove the dN/dS analyses (diminishing its findings), or go back and perform the dN/dS analysis using the genome data from multiple species presented in the 2021 GBE Wyka et al paper.

Specific comments:

line 86 - Wyka et al. (2020a) citation - besides updating this reference, make sure it also is listed as 2020a in the Reference list (not just 2000).

line 126 - Liu et al. Accepted - This did not appear in the list of references cited.

line 248 - conserved

lines 280-1 - is it possible that there is either bias or some difficulty in estimating the inferred recombination rate associated with duplicated genes (such as potential problems with read mapping and potential genotype errors for these genes)?

Line 299 - you state genes with conserved domains are most frequent within hotspots - but you are not indicating any overrepresentation, are you? Don't we expect these genes to be most frequent in hotspots because they are most frequent across the whole genome?

Line 359-60 - Reword, there is not literally a lack of recombination here.

Lines 447-449 - Your wording seems to indicate that recombination rate is determining whether purifying selection or positive selection is acting, but that is not true. Purifying and positive selection are not mutually exclusive across the genome (one can be happening at some loci, the other at other loci). And high recombination is expected to increase the efficiency of both types of selection. I don't see how high recombination could ever explain few signatures of positive selection.

Lines 508-9 - rephrase "likely controlled by" -> likely kept under control, or similar.

6. PLOS authors have the option to publish the peer review history of their article (what does this mean?). If published, this will include your full peer review and any attached files.

Reviewer #1: **Yes: **Zachary Konkel

Reviewer #2: No

Reviewer #3: **Yes: **Christopher Toomajian

---

## [Author Response · Author response to Decision Letter 0]

29 Jun 2021

We thank you for your feedback on our manuscript. We have addressed your comments and have

altered the text in accordance with your advice.

Reviewer #1:

Major concerns:

252-254: I would be more supportive about the evidence for accessory chromosome(s) if these

data were derived from the population-level accessory pangenome. In this study the accessory

pangenome references all isolates gathered across the globe – global populations may not

exchange genetic material, so 1) How can you distinguish between accessory chromosome

scaffolds and genomic regions that evolved from isolation? 2) How can you distinguish them

from poorly assembled scaffolds? Short reads are especially prone to not assembling well,

potentially leaving streaks of N’s in scaffolds derived from small contigs... particularly from

regions with low coverage. At this point, the study points to accessory chromosomes - my

assessment is that accessory chromosomes are not substantiated by these data and the other 2

options should be presented as the assumed nulls. – Claviceps purpurea is still spread globally

through import and export of contaminated grain. In fact, one of the isolates that we have

sequenced for this study that was from New Zealand was sent to us by an APHIS agent at a

check point in Oregon from an import of contaminated seed. It is very plausible to suggest that

not all contaminated seed is identified and that some foreign samples of C. purpurea do cross

country borders. In addition, Liu et al. 2020 (https://doi.org/10.1002/ece3.7028) identified that

geographical distribution did not correlate with genetically sub-divided populations of C.

purpurea. While this section represented a very small portion of our paper, we have further

reduced the discussion and presentation of these results and are simply stating a hypothesis that

will need to be studied with the use of long-read sequencing and a larger population of samples.

Other concerns:

639-646: I’m concerned about the type 1 error thresholds – considering the individual type 1

error thresholds, I think the cumulative type 1 error for the study could be large. The data seems

justified visually and corrections are in place in explicit multiple comparisons analyses, but the

statistical power of the overall study concerns

me. https://www.ncbi.nlm.nih.gov/pmc/articles/PMC4840791/ - We understand reviewer 1’s

concern regarding type 1 error. As noted by the reviewer we have made our best attempt to limit

the potential for type 1 error, and the statistical analyses demonstrate this. Given there are

multiple analyses in the manuscript it is not possible to measure the statistical power of the entire

study. Therefore, we believe we have provided an accurate interpretation of the data, and do not

try to over extrapolate these results in the discussion and know there is the potential for type 1

error. Only future experimental data will be able to determine the validity of some of our 

analyses

Use of normative language without reference points, e.g. vague adjectives as in line 45 “broad

genomic analyses”; general grammatic errors/tense switching/active-passive voice switching,

e.g. 89 “the factors that were influencing” – We have addressed this throughout the manuscript.

Abstract:

“which is often governed by variable gene content.” - what is governed by variable gene content,

what does this mean, what information does this add? - Statement has been removed.

Data locations:

“Dataset, doi: XXX” – remove or add doi – doi has been added

Author Summary:

43-44: “ambiguity surrounding the true nature” is vague, please be specific – Statement has been

made more succinct.

Introduction:

*88-89: how does tandem gene duplication implicate their positive contribution to Claviceps –

how is this definitively not just a random duplication? Looking for a citation here or explanation

of how – Statement has been reworded.

106: “use SCOs to identify genes under positive selection” – the SCOs don’t show which genes

are under positive selection, what was the actual method that references the SCOs? – Statement

has been altered

Results:

183-184: here, the percentage in ‘()’ is relative to the whole then the next is relative to the

previous percentage; previous orthogroup percentages in the text had a common reference point.

I think readers could infer, so I can see arguments either way, but I would argue either maintain

common reference points or explicitly state them so the reader doesn’t have to infer. – Changed

to be consistent with relative to the whole.

185: “two conserved genes” – do you mean orthogroups? – Yes, but these are single-copy

orthologs. We believe the designation of gene is appropriate.

224: again, I’m not sure I’d say “genes” when you are referencing what appear to be

orthogroups… If there are paralogs within these orthogroups then I don’t think it is appropriate

to universally asign a function to the OG because paralogs can take on different functions. At

present, it might be appropriate if they are single copy orthologs, but I don’t think that is made

clear if that’s the case. – It has been previously stated that the dN/dS analysis and positive 

selection analysis were only conducted on single-copy orthologs (no paralogs in the orthogroup).

Since we are only using single-copy orthologs, we have kept with using the term “gene”

252-254: see major concerns – See comments for major concerns.

*255-270: are these obtained from individual populations or the entire sample set? – Entire

sample set.

282: RIP is introduced without expanding the acronym. – Acronym has been expanded

Discussion:

*338-339: how is it supported that this is potentially due to a lack of RIP? – Statement has been

removed

417-420: this is not substantiated; perhaps instead of saying “it is plausible to believe”, state the

two hypotheses you have mentioned. At present, this is added without any support. - We have

altered the text and decided against stating the hypothesis regarding increased mutation due to

agricultural practices.

Conclusions:

503-504: This section would benefit from reducing normative language i.e. “large” is used twice,

“high” is used - there’s no reference to base these off. Use actual values and if you want to

compare with other fungi then actually some sort of reference – Normative language has been

toned down.

Methods:

639-646: see general concerns – See comments regarding statistics above.

Reviewer #2: PONE-D-21-02186

A large accessory genome, high recombination rates, and selection of secondary metabolite

genes help maintain global distribution and broad host range of the fungal plant pathogen

Claviceps purpurea

The authors conducted a comprehensive study on the extended gene repertoire (pangenome) of

Claviceps purpurea sampled from different hosts in six countries. The inclusion of samples from

different geographical locations and the total number of 24 genomes would be adequate for

exploring for the first time the pangenome of this pathogen. Overall, the results presented bring

novelty and would be fit for publication in PLOS One after major revisions. Major criticisms are

regarding long parts of the discussion being mostly speculations without robust results for 

supporting their claims. Therefore, the discussion needs to focus on the major novelties

presented (i.e., positive selection in secondary metabolites) and shortened in other sections (i.e.,

mini-chromosome speculation, several comparisons with Zymoseptoria tritici). In my detailed

comments below, I request some clarifications and suggest modifications in the manuscript.

The manuscript figures are of high quality, and the authors provide open access to scripts and

genomic data, which should be praised.

Specific comments organized according to line numbers in the manuscript:

Results

FigS1: What is the impact of the hosts of origin on the population structure? – This analysis is

not a focus of this paper but has been discussed in Miao Liu et al. 2020

(https://doi.org/10.1002/ece3.7028). In general, C. purpurea population genetic subdivisions are

not correlated with hosts of origin.

FigS1: Mention the enlarged view within the PCA. - Enlarged PCA has been mentioned.

Line 126: Liu et al not referenced. Would it be possible to provide preprint? – It has since been

published and is available https://doi.org/10.1002/ece3.7028. We have updated the reference

section.

Line 131: Is 59.2% correct? – Yes, 6,244 / 10,540 = 59.2%. Relative to the whole (pangenome)

Line 151: Please clarify what would be the consequences of unclassified orthologs in the

accessory genome for the GO enrichment. – An increased number of unclassified orthologs in

the population of the accessory genome will likely increase the type 2 error rate and may mask

some potentially enriched GO terms. This caveat has been stated.

Line 200: I understand the details for the “stringent filtering” are in the methods, but briefly

mention here. – We have deleted this section, due to removal of the positive selection analysis.

Line 220: Given that effectors in the core genome are not under selection in the present study,

how authors see the role of effectors in the accessory genome? – We have deleted this section,

due to removal of the positive selection analysis.

Line 252: Since authors used Illumina reads mapped on a not complete (chromosome level)

reference genome, how to distinguish whether different scaffolds would ultimately compose a

single chromosome and represent indels instead of mini-chromosomes? Also, what would be the

impact of structural variations in the pangenome (core + accessory + indels)? – Illumina reads

were not mapped to a reference genome. All genomes were assembled de novo in Wyka et al.

2021, therefore this comment is not pertinent to our study.

Line 283: Why have authors not considered DNA transposons? – We have added DNA

transposons to the analysis. The supplemental figure has been updated.

Line 298: “Some hotspots showed a greater association with duplicated genes and TEs (Fig. 7 BD), while others showed a lower association (Fig. 7 A, E).” ….. If I’m interpreting the figure

correctly, I’d suggest being more explicit for Figure 7A = no TE’s within the recombination

hotspot. – Text has been made more explicit to follow reviewer’s suggestion.

Discussion

As a general comment, please remove references to tables and figures. – We have received

feedback from the editor that is it our choice to remove/keep the references to figures in the

discussion. Due to the abundance of figures in the study and supplemental files, we feel that it

would improve the ability of the reader to continue to track our discussion with references back

to the pertinent figure / data.

Line 363 to 382 is mainly speculative as long reads would be necessary to tackle the minichromosome hypothesis. Would it be possible to remove/shorten this section? – This section has

been shortened, but we decided to leave it for the purpose of stating the hypothesis which could

be important for C. purpurea evolution if dispensable are identified by another researcher

through the use of long-read sequencing.

Line 324: Zymoseptoria tritici is not a biotrophic fungus (see

references: doi.org/10.1016/j.fgb.2015.04.001and
doi.org/10.1094/phi-i-2011-0407-01), and the

only grass it infects is wheat (Please

see http://www.genome.org/cgi/doi/10.1101/gr.118851.110 and also Seifbarghi et al 2009).

Please, authors should correct this statement and discussion based on this wrong definition.

Please clarify that the global distribution of Zymoseptoria tritici is due to wheat infection and

that grasses (other than wheat) are infected by different Zymoseptoria.

Line 325: Please tone down comparisons with Zymoseptoria tritici. Z. tritici is heterothallic, has

a very limited host range, and develops across a crop season mostly via asexual spores (and

hyphae). – (comments for Line 324 and Line 325) We have corrected the statements regarding

Z. tritici and have altered the discussion around these statements. However, regardless of the fact

that Z. triciti is only globally distributed because of wheat infections we still feel there is ample

claim to compare these two pathogens. Claviceps purpurea is also a pathogen of wheat as is

globally distributed (mostly due to the spread of contaminated seeds between farms). While C.

purpurea does have a larger host range, both C. purpurea and Z. tritici are under similar

ecological and environmental pressures to survive, which have been shown to influence similar

pangenome structures in bacteria.

Line 343: Is there RNAseq data available for improving gene annotation? Considering the 

signatures of positive selection on unclassified genes, how would positive selection act on

pseudogenes in C. purpurea? – The RNAseq data that was available at the time of assembly and

annotations (Wyka et al. 2021) was already used in the annotation pipeline to help improve gene

predictions. We did specifically look at pseudogenes, but the few pseudogenes that have been

found in C. purpurea are associated with secondary metabolites, and specifically ergot alkaloids.

Gene gains, gene loss and gene sequence changes at the end of chromosomes are believed to be

major drivers of ergot alkaloid diversification in the Claviceptaceae (Young et al. 2015). It is

possible that many of the many of the unclassified genes are actually psuedogenes that represent

remnants of gene loss that have given rise to diverse chemotypes (Young et al. 2015).

Line 428: Populations in areas without fungicide pressure or deployment of resistant cultivars

(Israel, Oregon) have similarly high levels of genetic diversity when compared to regions with

intensive usage and deployment (Switzerland, Oregon R) as for example in the study of

Hartmann et al. 2017. I suggest authors to shorten this discussion section to avoid speculations. –

We have removed this discussion point, and left the hypothesis of RIP “leakage”, which is likely

a more plausible hypothesis.

Line 435: Are C. purpurea of different hosts able to cross sexually? How would it influence

population recombination rates and hotspots analyses? – C. purpurea of different hosts are able

to cross sexually and do so quite often. This likely increases the effective population size of C.

purpurea which may inflate recombination rates. However, a larger-scale recombination analysis

comparing multiple fungal organisms with LDhot will likely be needed to determine how

different lifestyles can influence hotspot analysis. At the time of writing, I only saw Z. tritici

being analyzed with LDhot.

Line 446: Are high levels of recombination expected in a homothallic fungus? – Sexual

recombination has been found to frequently occur in homothallic plant pathogens, such as

Sclerotinia slcerotiorum (Attanayake et al. 2014, https://www.nature.com/articles/hdy201437)

and Fusarium graminearum (Talas & McDonald 2015,

https://bmcgenomics.biomedcentral.com/articles/10.1186/s12864-015-2166-0), which have large

effective population sizes. Certainly, more research needs to be done across additional

homothallic fungal species but recombination among homothallic species does appear to occur,

but the rate of recombination may depend on the population size and species.

Line 478: I see this section on the evolution of secondary metabolites in C. purpurea as a novel

and relevant discussion that should receive focus by shorting previous sections. – This section

has largely been removed, due to the removal of the positive selection analysis.

L 488: Reference about Puccinia missing. - The statement was removed from the text due to

removal of the CodeML analysis, so no reference is needed.

Materials and Methods

Table 1: I could not find LM28 and 582 in the PRJNA528707. – These genomes are present in

PRJNA528707 and have recently been uploaded by NCBI, please see the screenshot of NCBI

below.

Line 522: Mention the sequencing technology for the 23 isolates. - Sequencing technology has

been mentioned.

Line 524: “cutoff of 50 aa” What does it mean? - Removed “aa” as it’s redundant.

Line 546: What RepBase version? - v24.03

Line 567: Describe how gene sequences of individual genomes were extracted for selection

analyses. – Single-copy orthologs from the pangenome analyses (described in the previous

section) were used for dN/dS analysis. This has been made more explicit.

Reviewer #3:

The authors use resequenced, assembled, and annotated genomes from 24 C. purpurea isolates to

perform a pangenome analysis, and combine this work with two additional analyses, estimating

recombination rates (and detecting hotspots), and performing an analysis of selection on coding

sequences (single copy orthologous genes). There are many different approaches to infer the action

of natural selection in population genetics datasets, and they authors here estimate selection

(purifying and positive selection) through estimating dN/dS (omega) for their population set.

The work presented here involves a fairly large dataset, as well as lots of bioinformatics analysis,

and so those contributions alone can serve as important resources for understudied species. That

said, these large-scale genomic analyses are not that well integrated in this manuscript (at least

they are integrated to some extent, especially through various enrichment analyses), and other

population genetic analyses could have been included that might help to integrate the different 

pieces in other ways. The interpretation of population genomic studies is not straightforward, and

results tend to be consistent with multiple explanations until more specific analyses are carried

out. In this work, the results tend to be over-interpreted, and in many cases the evidence for the

authors' conclusions are rather weak. However, unless some important methods descriptions have

been left out of this manuscript, the biggest problem in the manuscript is the analysis of natural

selection. It appears the authors have carried out the dN/dS analyses with PAML and the CodeML

algorithm using only the 24 C. purpurea isolates and without inclusion of data from additional

species. The Jeffares et al. paper they cite indicates sequences from multiple species are needed

for these analyses [states that PAML FAQ recommends a minimum 4-5 species, refers to multiple

alignments of protein-coding gene sequences from several species in a phylogeny, in describing

required input files indicates an annotated genome for at least 1 related species is needed besides

the target species of interest]. dN/dS analyses are meant to compare nonsynonymous versus

synonymous substitutions (alleles gone to fixation in some species or independent lineages), yet

what we have in a populations capable of sexual reproduction are segregating polymorphisms

(since the isolates are not independent lineages), and the distinction is important. Kryazhimskiy

and Plotkin (https://www.ncbi.nlm.nih.gov/pmc/articles/PMC2596312/) have shown that inferring

selection pressures from dN/dS calculated from population samples does not work in the same way

that dN/dS analyses work when multiple species are compared (the behavior of dN/dS is expected

to be different). In particular, they find that even for genes experiencing negative selection, the

observation of an elevated dN/dS (values closer to 1) is expected with intra-specific samples. And

the observation of dN/dS<1 is also consistent with strong positive selection. Given this problem,

the major report from the manuscript that secondary metabolism genes are more important than

effectors based on the dN/dS analyses must be called into question, since it no longer has any

reliable support. – We have incorporated all Claviceps species from the Wyka et al. 2021 paper

and attempted to re-run the analysis on 53 genomes. With the resources on hand we were able to

conduct a re-analysis of dN/dS (omega) ratios with YN00, however, the use of the site models in

CodeML were too computationally intensive (~24+ hours per single-copy ortholog, ~4,000 in the

new dataset) for our current computing capacity. This analysis would require a large expense and

time that we currently do not have. Both the lead author (Wyka) and corresponding author

(Broders) are no longer at Colorado State University (where the server for the originally analyses

was located) and have moved on to different positions respectively, and no longer have access to

the CSU server. We do not think the lack of this analysis diminishes the results of this paper or

lessens the value of the findings.

The selection analyses represent a major section of the manuscript - lines 174-238 in the Results,

566-588 in the Methods, Figures 3 and 4, Table 2, Supplemental figures S3, S5, S9, and

supplemental tables S4-S7 all are based on the highly questionable dN/dS results. To become

publishable, the manuscript should either remove the dN/dS analyses (diminishing its findings),

or go back and perform the dN/dS analysis using the genome data from multiple species

presented in the 2021 GBE Wyka et al paper. – As mentioned above, we have re-done the dN/dS

using the Claviceps genomes, but have removed the CodeML positive selection analysis.

Specific comments:

line 86 - Wyka et al. (2020a) citation - besides updating this reference, make sure it also is listed

as 2020a in the Reference list (not just 2000). – Changes have been made.

line 126 - Liu et al. Accepted - This did not appear in the list of references cited. – Correct

reference has been added

line 248 – conserved - Done

lines 280-1 - is it possible that there is either bias or some difficulty in estimating the inferred

recombination rate associated with duplicated genes (such as potential problems with read

mapping and potential genotype errors for these genes)? – For our recombination analysis we

used a method that utilizes haploid whole genome alignments with LastZ and MultiZ

(Stukenbrock and Dutheil 2018a), so there should be no concerns of errors with read mapping or

genotype errors. Unless I am mistaken.

Line 299 - you state genes with conserved domains are most frequent within hotspots - but you

are not indicating any overrepresentation, are you? Don't we expect these genes to be most

frequent in hotspots because they are most frequent across the whole genome? – Yes, this may

be expected, however, we did not compute any enrichment analysis as we only observed 5

recombination hotspots and felt that the statistical power of this test would be very low.

Line 359-60 - Reword, there is not literally a lack of recombination here. - The statement has

been reworded.

Lines 447-449 - Your wording seems to indicate that recombination rate is determining whether

purifying selection or positive selection is acting, but that is not true. Purifying and positive

selection are not mutually exclusive across the genome (one can be happening at some loci, the

other at other loci). And high recombination is expected to increase the efficiency of both types

of selection. I don't see how high recombination could ever explain few signatures of positive

selection. - We agree with your comment, we have altered the text to provide clarity to the

statement.

Lines 508-9 - rephrase "likely controlled by" -> likely kept under control, or similar. – Changed

to “kept under controlled”.

---

## [Decision Letter · Decision Letter 1]

17 Aug 2021

PONE-D-21-02186R1

A large accessory genome and high recombination rates help maintain global distribution and broad host range of the fungal plant pathogen Claviceps purpurea

PLOS ONE

Dear Dr. Broders,

Thank you for submitting your manuscript to PLOS ONE. After careful consideration, we feel that it has merit but does not fully meet PLOS ONE’s publication criteria as it currently stands. Therefore, we invite you to submit a revised version of the manuscript that addresses the points raised during the review process.

I appreciate the changes that you have made in the revision, and although I have decided Minor Revisions are required, they may be relatively simple to make.  First, in addition to providing my own review on the revision (Reviewer 3) and seeking the comments of one of the original reviewers (Reviewer 2), I sought the opinion of a new reviewer for the resubmission (Reviewer 4) since the revision is substantially different from the original submission with the removal of major sections.

My comments emphasize some needed changes in light of the updates of the selection analyses, but these changes are simply in wording. Following that general theme, I have also suggested other wording changes where statements or conclusions in the manuscript are not strongly supported by analyses (such as, where other alternative factors could explain a finding instead of, or in combination with a factor you highlight). These changes ought to help the reader not as fluent in the analyses you have performed distinguish some opinions or hypotheses you are proposing from results that are very strongly supported by the data and analyses.

I very much appreciate the insights of Reviewer 4. The PLOS ONE policy is to accept publications regardless of perceived impact, and so for a genomics manuscript like this, a reviewer may suggest insightful analyses to include but these additions often cannot be required when their inclusion would only increase the work's impact. Reviewer 4 states as much, indicating the concerns raised shouldn't stop the acceptance of the manuscript as long as sufficient explanations can be provided for why suggested analyses cannot be performed (or are too impractical for the authors). That said, I strongly recommend you try to follow through with at least some of these new suggestions

[e.g., is it feasible to take accessory genes for which orthologous sequences are available in the other species to compute dN/dS? This would allow for some interesting comparisons on the nature of selection in the core and accessory genomes. Would you agree that creating some summary and/or figures relevant to the genomic distribution or density of core, accessory, and singleton genes along the reference genome scaffolds could be informative for the structure of the accessory genome and the potential for accessory chromosomes? The suggestion related to host species is interesting too, though it may be that not enough isolates share host species to make this analysis meaningful.]

Other comments emphasize unclear sections, especially related to reference scaffolds missing from the multiple alignments. I know full details are available in cited references, but providing some quick explanations for how procedures from those cited references can result in 'missing scaffolds' in this work is critical for interpreting and conceptualizing the results. I recommend that when addressing comments, some edits, even if short, be made in the manuscript (rather than simply providing a clarification as a response to reviewers that readers may not be likely to see).

For any clarifications, please do not hesitate to contact me.

We look forward to receiving your revised manuscript.

Kind regards,

Christopher Toomajian

Academic Editor

PLOS ONE

Journal Requirements:

Reviewers' comments:

Reviewer's Responses to Questions

**Comments to the Author**

1. If the authors have adequately addressed your comments raised in a previous round of review and you feel that this manuscript is now acceptable for publication, you may indicate that here to bypass the “Comments to the Author” section, enter your conflict of interest statement in the “Confidential to Editor” section, and submit your "Accept" recommendation.

Reviewer #2: All comments have been addressed

Reviewer #3: (No Response)

Reviewer #4: (No Response)

2. Is the manuscript technically sound, and do the data support the conclusions?

Reviewer #2: Yes

Reviewer #3: Partly

Reviewer #4: Yes

3. Has the statistical analysis been performed appropriately and rigorously? 

Reviewer #2: Yes

Reviewer #3: Yes

Reviewer #4: Yes

4. Have the authors made all data underlying the findings in their manuscript fully available?

Reviewer #2: Yes

Reviewer #3: Yes

Reviewer #4: Yes

5. Is the manuscript presented in an intelligible fashion and written in standard English?

Reviewer #2: Yes

Reviewer #3: Yes

Reviewer #4: Yes

6. Review Comments to the Author

Reviewer #2: PONE-D-21-02186R1

A large accessory genome and high recombination rates help maintain global distribution and broad host range of the fungal plant pathogen Claviceps purpurea

The authors have addressed all my comments with clear and satisfactory answers. Also, they significantly improved the manuscript by extensively removing and re-writing several parts. After careful reading of this new version, I can recommend it for publication in PLOS ONE.

Reviewer #3: The authors have recomputed the dN/dS ratios for their set of gene orthologs, incorporating sequences from species besides C. purpurea, and have removed the analysis of positive selection based on CodeML site models. Those changes have satisfied the main problem I had identified in the original submission. Those edits have a substantial impact on the overall message of the manuscript, and having gone through the revisions, I have identified some other issues that require changes, detailed below.

Tone of manuscript and amount of evidence for conclusions. The strengths of the manuscript come from its scale and breadth (i.e., pan genome, genome-wide, multiple population genomics analyses). However, that breadth comes at a cost of identifying some interesting correlations between population genome features but not being able to demonstrate any strong evidence for causation in these cases. This explains my choice of Partly to answer the question Do the data support the conclusions? There are several general conclusions or statements in the manuscript that suggest causation but are really not well supported by the data and analyses, and these should be revised to tone them down.

Title - Factors X and Y “help maintain global distribution and broad host range”

Abstract – L26 “likely maintained by high recombination… ”, L28 “likely controlled by frequent recombination”

Intro L98 “likely maintained by”

Certain references to genes under positive selection – see comment below.

Appreciation for low power of whole-gene dN/dS ratio analysis. I have 2 points related to the interpretation of the whole gene dN/dS ratios. 1st) Genes with dN/dS >1 are not necessarily under positive selection, particularly for genes with ratios only slightly greater than 1. Some test would be required to show a particular ratio is significantly greater than 1. It would be better to refer to genes with the ratio >1 as positive selection candidates, or genes with signatures (or evidence) of positive selection (phrases that are used in several places in the manuscript). The following phrases should be revised: L96 “identified genes under positive selection,” L115 “landscape of genes under positive selection”. 2nd) A dN/dS ratio for a whole gene being > 1 is a very stringent criterion for positive selection (that is, at least when it is significantly greater than 1). The literature is full of examples where positive selection is indicated but where the whole gene dN/dS ratio is < 1. That means this analysis has very low power for detected genes with sites under positive selection, and so the absence for evidence of positive selection for a category of genes should not be confused with evidence for the absence of positive selection. When considering factors that can explain no evidence for positive selection on a subset of genes (such as L489-490), the most important one should be this lack of power to detect it.

Keep in-text reference citations consistent (all numerical, without publication year). As appropriate, most reference citations have been converted to numerical format, but some hold-overs of the (author, year) style remain. For example, L118, “in Wyka et al. (2021)” should be edited to “in Wyka et al. [17]” or even just “in [17].”

Minor points:

L112 should also reference [17]

L114 Amino acid cutoff requires some context here to be clear (such as the phrase For the purpose of defining gene models for subsequent analysis…). The appropriate context can only be found in the referenced papers currently.

L162 After filtering for positive selection – this analysis has been removed from the manuscript, correct? Then this sentence needs to go as well.

Gap between L203-204. Something is missing between hotspots and 1000 simulations.

L379 selection of secondary metabolite genes – this result is gone from the manuscript, so this needs to be removed here as well.

L400 Citation of Fig S7 should be changed to Fig S5.

Table S6 lists 36 scaffolds, but the text refers to 37 missing scaffolds.

Reviewer #4: This study explores the pangenome of Claviceps purpurea, as well as the recombination and selection landscapes based on 24 isolates. It is overall a nice study with interesting new results. I understand that this is a revised version of a previous manuscript, and, although I was not part of the previous reviewing process, it seems that the major methodology flaws pointed out were corrected by the authors. A sizeable part of the original analyses has been left out of the current manuscript entirely as the authors state that they do not have access to the computational power they had when the study was first written. As such I have been careful to make only suggestions for which the data should already be generated.

The introduction is clear and sets up the following text well. The methods section is detailed and scripts are provided publicly. The results are clearly explained. I have some concerns with the discussion which is not yet at a level of clarity as excellent as the rest of the manuscript. I also have some questions regarding the pangenome-related analyses.

Main comments:

One concern with the manuscript as it stands now is that the different sections feel very disconnected from each other, both in the results and in the discussion. The study starts with a description of the pangenome. Then the authors go on looking at selection only on core genes and only look at the recombination, once again, in fragments shared by all samples. I did not find an explanation in the manuscript as to why accessory genes found at a sufficiently high frequency could not be analysed for selection for instance and the results between accessory and core genes then compared.

Many of the comparisons made to other species are made with Z. triciti which has an accessory genome organized in core chromosomes separated from accessory chromosomes. Some of the comparisons would be more relevant if there were hints of a similar organization in C. purpurea. Here, despite the authors basing their study on de novo assemblies of the genomes, the only attempt to understand the distribution in the genome of core and accessory genes is a comment on fragments that are “missing” from the MGA. It is not clear to me what “missing” implies. Does this mean that the fragments are found in only one isolate and that these are not reported by the alignment tool? Are these the contigs containing the singleton genes? If so, the numbers do not match. Where are the other singleton genes? If they are aligning, would this not point to a possible issue with the way the pangenome was defined in the first place? Please clarify this as this could potentially be a problem.

The authors hypothesize that these missing contigs are a hint of possible accessory chromosomes. I (like previous reviewers) am not at all convinced by this argument. Indeed, the abundance of reverse transcriptase and other repeat associated genes on these fragments would point more towards assembly artefacts.

However, I do believe that understanding the genomic distribution of the core and accessory regions is extremely relevant to the topic of this study and should be attempted. Where are the accessory genes in the de novo assemblies? Are accessory/singleton genes systematically found together on the same contigs? Are they found more often on the same contigs or do they share contigs with core genes? I believe that the data should already be generated by the authors to answer most of these questions without need for new in-depth analyses.

Accessory genomes are frequently hypothesized (and indeed in some cases proven) to play a role in host specificity. I’m surprised that no mention of this was made in the results (perhaps I missed it?). Several of the strains come from the same host species (Hordeum vulgare, for example). Were there accessory genes shared by these strains? Surely, the data is already available easily from the orthogroups table. This also seems like a relevant trait to look for when describing the pangenome of a species with a broad host range, especially as the authors report an enrichment of effectors in the accessory genes.

None of these concerns are a cause for stopping the publication of the manuscript even without any new analyses, if appropriate explanations are provided as to why these analyses would be impossible/not relevant to the study.

Minor comments

As previously noted by the reviewers the discussion contains large section of comparison to Z.tritici and these are not always so relevant that they deserve the word count they get.

L.387: The authors argue that the host distribution of these two species are similar. Z.tritici is strictly a wheat pathogen when the authors state that C. purpurea “has a broad host range of ~ 400 grass species across 8 grass tribes”. Arguing that the hosts are similar enough to lead to a convergent evolution of pangenome size seems to be a stretch.

L.407. “Badet et al. suggested that TEs were likely contributing to Z. tritici accessory genome“. In this study, the accessory chromosomes are including non-genic content (based on PacBio genome assemblies), making the TEs having a clear role in expansion of the accessory genome. I am not sure how relevant this comparison is to a pangenome in which only the genes were considered. Perhaps focus this paragraph more on the results obtained in the current study instead of describing in details results from another paper.

L. 403. “While this analysis was only conducted on single-copy core genes, it suggests that some of the unclassified accessory genes (Fig 2 H) may be undergoing similar evolutionary trends.” What exactly is suggesting such a thing? See main comment regarding this.

L. 428. “due to our Illumina based we did not process“ Missing word? Assemblies?

L. 435-444. This paragraph is about TEs, genome size and gene duplication. The previous one is not, but the one even before is. This makes the reading confusing. Perhaps merge the two related paragraphs together.

L.442. Typo: “Clavicpes “

7. PLOS authors have the option to publish the peer review history of their article (what does this mean?). If published, this will include your full peer review and any attached files.

Reviewer #2: No

Reviewer #3: **Yes: **Christopher Toomajian

Reviewer #4: No

---

## [Author Response · Author response to Decision Letter 1]

23 Sep 2021

We thank you for your feedback on our manuscript. We have reviewed your comments and suggested edits and have done our best to address these comments. We have made a number of edits that we believe take into consideration your advice and address the concerns of reviewers. 

Reviewer #2: PONE-D-21-02186R1

A large accessory genome and high recombination rates help maintain global distribution and broad host range of the fungal plant pathogen Claviceps purpurea

The authors have addressed all my comments with clear and satisfactory answers. Also, they significantly improved the manuscript by extensively removing and re-writing several parts. After careful reading of this new version, I can recommend it for publication in PLOS ONE. 

Thank you for you feedback.

Reviewer #3: The authors have recomputed the dN/dS ratios for their set of gene orthologs, incorporating sequences from species besides C. purpurea, and have removed the analysis of positive selection based on CodeML site models. Those changes have satisfied the main problem I had identified in the original submission. Those edits have a substantial impact on the overall message of the manuscript, and having gone through the revisions, I have identified some other issues that require changes, detailed below.

Tone of manuscript and amount of evidence for conclusions. The strengths of the manuscript come from its scale and breadth (i.e., pan genome, genome-wide, multiple population genomics analyses). However, that breadth comes at a cost of identifying some interesting correlations between population genome features but not being able to demonstrate any strong evidence for causation in these cases. This explains my choice of Partly to answer the question Do the data support the conclusions? There are several general conclusions or statements in the manuscript that suggest causation but are really not well supported by the data and analyses, and these should be revised to tone them down.

Title - Factors X and Y “help maintain global distribution and broad host range” – The title has been altered

Abstract – L26 “likely maintained by high recombination… ”, L28 “likely controlled by frequent recombination” - These statements have been toned down

Intro L98 “likely maintained by” – This statement has been toned down

Certain references to genes under positive selection – see comment below.

Appreciation for low power of whole-gene dN/dS ratio analysis. I have 2 points related to the interpretation of the whole gene dN/dS ratios. 1st) Genes with dN/dS >1 are not necessarily under positive selection, particularly for genes with ratios only slightly greater than 1. Some test would be required to show a particular ratio is significantly greater than 1. It would be better to refer to genes with the ratio >1 as positive selection candidates, or genes with signatures (or evidence) of positive selection (phrases that are used in several places in the manuscript). 

The following phrases should be revised: 

L96 “identified genes under positive selection,” – This phrase has been re-worded.

L115 “landscape of genes under positive selection”. – This phrase has been re-worded.

2nd) A dN/dS ratio for a whole gene being > 1 is a very stringent criterion for positive selection (that is, at least when it is significantly greater than 1). The literature is full of examples where positive selection is indicated but where the whole gene dN/dS ratio is < 1. That means this analysis has very low power for detected genes with sites under positive selection, and so the absence for evidence of positive selection for a category of genes should not be confused with evidence for the absence of positive selection. When considering factors that can explain no evidence for positive selection on a subset of genes (such as L489-490), the most important one should be this lack of power to detect it. The reviewers make an excellent point and he have incorporated this into the text. – New lines 504-506

Keep in-text reference citations consistent (all numerical, without publication year). As appropriate, most reference citations have been converted to numerical format, but some hold-overs of the (author, year) style remain. For example, L118, “in Wyka et al. (2021)” should be edited to “in Wyka et al. [17]” or even just “in [17].” – We have altered the references to be more consistent. If a reference was at the start of a sentence we decided to go with Wyka et al. [17], for all other cases we used the numerical format.

Minor points:

L112 should also reference [17] – Reference has been added.

L114 Amino acid cutoff requires some context here to be clear (such as the phrase For the purpose of defining gene models for subsequent analysis…). The appropriate context can only be found in the referenced papers currently. – We have altered the sentence to reflect the reviewers feedback.

L162 After filtering for positive selection – this analysis has been removed from the manuscript, correct? Then this sentence needs to go as well. – Sentence has been removed

Gap between L203-204. Something is missing between hotspots and 1000 simulations. – The sentence has been corrected.

L379 selection of secondary metabolite genes – this result is gone from the manuscript, so this needs to be removed here as well. – The statement has been removed

L400 Citation of Fig S7 should be changed to Fig S5. – Has been changed

Table S6 lists 36 scaffolds, but the text refers to 37 missing scaffolds. – Has been corrected to 36 in the text.

Reviewer #4: This study explores the pangenome of Claviceps purpurea, as well as the recombination and selection landscapes based on 24 isolates. It is overall a nice study with interesting new results. I understand that this is a revised version of a previous manuscript, and, although I was not part of the previous reviewing process, it seems that the major methodology flaws pointed out were corrected by the authors. A sizeable part of the original analyses has been left out of the current manuscript entirely as the authors state that they do not have access to the computational power they had when the study was first written. As such I have been careful to make only suggestions for which the data should already be generated.

The introduction is clear and sets up the following text well. The methods section is detailed and scripts are provided publicly. The results are clearly explained. I have some concerns with the discussion which is not yet at a level of clarity as excellent as the rest of the manuscript. I also have some questions regarding the pangenome-related analyses.

Main comments:

One concern with the manuscript as it stands now is that the different sections feel very disconnected from each other, both in the results and in the discussion. The study starts with a description of the pangenome. Then the authors go on looking at selection only on core genes and only look at the recombination, once again, in fragments shared by all samples. I did not find an explanation in the manuscript as to why accessory genes found at a sufficiently high frequency could not be analysed for selection for instance and the results between accessory and core genes then compared. – We provided an additional rebuttal to this comment about accessory genes in the following concerns. To answer the question, “why we didn’t include accessory genes in the selection analysis”, we chose not to do so as previously we only included C. purpurea isolates for the dN/dS ratio. This approach was incorrect, so we needed to include other species to provide power to the selection analysis. We did do this by including the remaining species that we had genomes for in the Claviceps genus. Choosing to run the selection on only core genes make sense as the dataset will be consistent, and all genes will have been compared to all other Claviceps species. If we started to look at accessory genes then we would also reduce the number of Claviceps species used in each dN/dS calculation. For example, core genes used alignments from 29 other Claviceps species to calculate dN/dS, however, if we start to look at accessory genes then the number of other Claviceps species will be reduced and thus would not provide a consistent dataset to compare against the core genes. 

Many of the comparisons made to other species are made with Z. triciti which has an accessory genome organized in core chromosomes separated from accessory chromosomes. Some of the comparisons would be more relevant if there were hints of a similar organization in C. purpurea. Here, despite the authors basing their study on de novo assemblies of the genomes, the only attempt to understand the distribution in the genome of core and accessory genes is a comment on fragments that are “missing” from the MGA. It is not clear to me what “missing” implies. Does this mean that the fragments are found in only one isolate and that these are not reported by the alignment tool? Are these the contigs containing the singleton genes? If so, the numbers do not match. Where are the other singleton genes? If they are aligning, would this not point to a possible issue with the way the pangenome was defined in the first place? Please clarify this as this could potentially be a problem. – It is difficult to follow the reviewers statement here as there are no direct references and they do not indicate which “numbers do not match up”. From the Multiple Genome Alignment (MGA) we only kept scaffolds (based on the reference genome) that had alignments from all of the remaining 23 genomes. The reference genome contained 191 scaffold and from the final MGA we kept 154 scaffolds. The scaffolds that were not kept contain mostly accessory and singleton genes (as stated in S6_table). However, there are some cases of core genes being present on these scaffolds. The next question will be “Then how are there core genes on scaffolds that are not part of the scaffolds that were kept”. The answer to this question is that the pangenome was conducted with OrthoFinder (as other pangenome papers have also used), which groups the proteins into orthologous groups (OGs). OGs are then classified as “Core” if they contain proteins from all 24 genomes, regardless of whether the OG is paralogous (i.e. some genomes may contain 2 copies of the gene while others contain 1). If the core OG is paralogous for the reference genome or all but 1 genome, then it is entirely possible that the 1+ genomes that contain only 1 copy of the protein will not have a DNA region to align to the scaffold that contains the 2nd copy. This scaffold would then lack all 24 genomes and not be kept from the MGA. Since only a handful of scaffolds were not kept, and contain a large majority of accessory genes and TEs we did not see any fault in the logic of how the pangenome was created. 

As for other accessory and singleton genes, the MGA also contain alignments that were either shared by 1-23 of the other genomes and not the reference. Therefore, these alignments were not kept as they did not map to the reference. 

The authors hypothesize that these missing contigs are a hint of possible accessory chromosomes. I (like previous reviewers) am not at all convinced by this argument. Indeed, the abundance of reverse transcriptase and other repeat associated genes on these fragments would point more towards assembly artefacts. – We are stating this more as a potential hypothesis, but have made it more clear in the text that assembly artifacts could also be a likely hypothesis.

However, I do believe that understanding the genomic distribution of the core and accessory regions is extremely relevant to the topic of this study and should be attempted. Where are the accessory genes in the de novo assemblies? Are accessory/singleton genes systematically found together on the same contigs? Are they found more often on the same contigs or do they share contigs with core genes? I believe that the data should already be generated by the authors to answer most of these questions without need for new in-depth analyses. – If we had chromosome level genomes I could understand the reviewers comment here about understanding where accessory genes are located, in comparison to core genes. However, without such data we do not believe the analysis would strengthen the paper, but in fact would provide a discussion on results that do not draw any strong conclusions. Since many of the de novo assemblies contained > 1,000 contigs we feel that such an analysis may also lead to inaccurate results as a contig that contains many accessory genes may in fact be flanked by two contigs that contains large amounts of core genes. Without long-read sequencing data we would not be able to accurately examine the true distribution of these genes.

Accessory genomes are frequently hypothesized (and indeed in some cases proven) to play a role in host specificity. I’m surprised that no mention of this was made in the results (perhaps I missed it?). Several of the strains come from the same host species (Hordeum vulgare, for example). Were there accessory genes shared by these strains? Surely, the data is already available easily from the orthogroups table. This also seems like a relevant trait to look for when describing the pangenome of a species with a broad host range, especially as the authors report an enrichment of effectors in the accessory genes. – Single isolates of Claviceps purpurea can infect multiple hosts and this has been shown in several different manuscripts over the years. While many isolates may have been isolated from a single host species, we do not believe that shared genes between that snap-shot of isolation will reveal any correlation to an ability to infect H. vulgare (for example). In addition, additional work that was sparked from this paper looked at population structure of C. purpurea and found that host groups did not show any correlation with population structures (DOI: 10.1002/ece3.7028). To really test this proposed hypothesis we would need to conduct a broad grass inoculation panel of all the C. purpurea strains used in this study to truly identify any lack of infection potential and then conduct a correlation analysis of gene content. Currently, we do not have the means and capability to conduct such inoculation trials as I have graduated and do not have funding / resources.

None of these concerns are a cause for stopping the publication of the manuscript even without any new analyses, if appropriate explanations are provided as to why these analyses would be impossible/not relevant to the study.

Minor comments:

As previously noted by the reviewers the discussion contains large section of comparison to Z.tritici and these are not always so relevant that they deserve the word count they get.

L.387: The authors argue that the host distribution of these two species are similar. Z.tritici is strictly a wheat pathogen when the authors state that C. purpurea “has a broad host range of ~ 400 grass species across 8 grass tribes”. Arguing that the hosts are similar enough to lead to a convergent evolution of pangenome size seems to be a stretch. – The statement has been altered to only reference the similar geographical distribution as the factor that may help explain a similarly large accessory genome.

L.407. “Badet et al. suggested that TEs were likely contributing to Z. tritici accessory genome“. In this study, the accessory chromosomes are including non-genic content (based on PacBio genome assemblies), making the TEs having a clear role in expansion of the accessory genome. I am not sure how relevant this comparison is to a pangenome in which only the genes were considered. Perhaps focus this paragraph more on the results obtained in the current study instead of describing in details results from another paper. – From the previous revision we have removed some of the discussion around Z. tritici, however, we have decided to leave this paragraph as it does provide a medium and helps convey the results we obtained from C. purpurea, as it provided a reference point for comparison with another organism.

L. 403. “While this analysis was only conducted on single-copy core genes, it suggests that some of the unclassified accessory genes (Fig 2 H) may be undergoing similar evolutionary trends.” What exactly is suggesting such a thing? See main comment regarding this. – This sentence has been removed as we did not conduct the analyses to support it’s message.

L. 428. “due to our Illumina based we did not process“ Missing word? Assemblies? – Yes, assemblies was missing.

L. 435-444. This paragraph is about TEs, genome size and gene duplication. The previous one is not, but the one even before is. This makes the reading confusing. Perhaps merge the two related paragraphs together. – We have altered the paragraph orientation to in agreement with the reviewers’ suggestions.

L.442. Typo: “Clavicpes “ - Fixed

---

## [Decision Letter · Decision Letter 2]

11 Oct 2021

PONE-D-21-02186R2A large accessory genome and high recombination rates may influence global distribution and broad host range of the fungal plant pathogen Claviceps purpureaPLOS ONE

Dear Dr. Broders,

Thank you for submitting your manuscript to PLOS ONE. After careful consideration, we feel that it has merit but does not fully meet PLOS ONE’s publication criteria as it currently stands. Therefore, we invite you to submit a revised version of the manuscript that addresses the points raised during the review process.

Consider this a conditional acceptance. Reviewer #4 has found the revisions acceptable, and only points out that the reviewers should deposit their data in Dryad and update the doi by the time of publication. Reviewer #3 (myself) only found one minor issue of substance -  making sure the number of "missing scaffolds" reported in different parts of the paper are all consistent. Beyond that, I discovered a handful of grammatical errors that need to be corrected (and regret that these were not caught in an earlier version). Since PLOS ONE does not copyedit accepted manuscripts, I would urge the authors to take this opportunity to check the whole of the manuscript carefully for any other spelling or grammatical errors that have been missed. The manuscript can be accepted without sending it out to reviewers once the authors have fixed the specific problems and errors listed in this round of review.

We look forward to receiving your revised manuscript.

Kind regards,

Christopher Toomajian

Academic Editor

PLOS ONE

Journal Requirements:

Reviewers' comments:

Reviewer's Responses to Questions

**Comments to the Author**

1. If the authors have adequately addressed your comments raised in a previous round of review and you feel that this manuscript is now acceptable for publication, you may indicate that here to bypass the “Comments to the Author” section, enter your conflict of interest statement in the “Confidential to Editor” section, and submit your "Accept" recommendation.

Reviewer #3: (No Response)

Reviewer #4: All comments have been addressed

2. Is the manuscript technically sound, and do the data support the conclusions?

Reviewer #3: Yes

Reviewer #4: Yes

3. Has the statistical analysis been performed appropriately and rigorously? 

Reviewer #3: Yes

Reviewer #4: N/A

4. Have the authors made all data underlying the findings in their manuscript fully available?

Reviewer #3: Yes

Reviewer #4: Yes

5. Is the manuscript presented in an intelligible fashion and written in standard English?

Reviewer #3: No

Reviewer #4: Yes

6. Review Comments to the Author

Reviewer #3: The authors have addressed nearly all of my comments from the previous round of review.

One issue I noted was the reported number of "missing scaffolds." The supplemental table lists 36, and the authors have changed the text in 1 place to agree with this, but another place in the text (line 396) the manuscript still lists 37 (and oddly, in their response to reviews, they explain that the reference genome had 191 scaffolds and 154 were kept in the final MGA, which would suggest 37 missing, unless this description is in error). This issue still needs to be resolved completely.

Beyond this, the following minor grammatical issues remain and should be corrected (as PLOS ONE does not copyedit).

Line 399 conversed domains -> conserved domains

Line 403 it still cannot be rule out -> it still cannot be ruled out

Line 403 may be assembly artifact -> either may be an assembly artifact OR may be assembly artifacts

Line 405 these are an important aspects -> these are important aspects OR these are an important aspect

Line 488 lack of power to positive -> lack of power to detect positive

Reviewer #4: The authors answer to my comments were satisfactory. I am happy to recommend the current version of the manuscript for publication in Plos One. (I did notice that the DOI for the dryad dataset in the main text does not lead to any dataset, but I am sure this would have been fixed by the authors at the time of publication.)

7. PLOS authors have the option to publish the peer review history of their article (what does this mean?). If published, this will include your full peer review and any attached files.

Reviewer #3: **Yes: **Christopher Toomajian

Reviewer #4: No

---

## [Author Response · Author response to Decision Letter 2]

9 Nov 2021

We thank the reviewers for their time and input. We have edited the manuscript one more time for grammar and incorporated the recommendations of the reviewers. We reviewed the data again and determined there are 37 missing scaffolds. One of the scaffolds doesn't contain any genes, so we likely did not picked it up before. We added contig 181 to S6_Table and fixed the 36's back to 37's in the text. The Dryad data has been submitted and is in progress and will be available before publication.

---

## [Editor Report · Decision Letter 3]

11 Nov 2021

PONE-D-21-02186R3A large accessory genome and high recombination rates may influence global distribution and broad host range of the fungal plant pathogen Claviceps purpureaPLOS ONE

Dear Dr. Broders,

Thank you for submitting your manuscript to PLOS ONE. After careful consideration, we feel that it has merit but does not fully meet PLOS ONE’s publication criteria as it currently stands. Therefore, we invite you to submit a revised version of the manuscript that addresses the points raised during the review process.

In making grammatical corrections, the authors have deleted 3 sentences in the Discussion (L485) that originally had been added in Revision 2 to address a point I had raised (as Reviewer 3 of Revision 1) – that the detection of natural selection in specific genes using a whole-gene dN/dS ratio of >1 as a criterion was a low power method, that in many cases natural selection may have acted but not caused this ratio to exceed 1. This deletion is not shown in the track-changes version of the manuscript, but can be seen by comparing this revision to the last. Please restore these sentences or something revised along the same lines, to make clear to the readers that the low power of the approach is likely a main reason that evidence of selection was detected only for a few genes (and no predicted effectors).

Additionally, the following grammatical errors (new and old) should be fixed at the same time:

L20 – subject-verb agreement. Change back to **need**. The subject is **inferences**, so the verb should remain plural.

L307 – Verb tense. contained -> contain

L416 – insert a ‘the’? – TEs were likely contributing to **the**
*Z. tritici *accessory genome

L785  - spelling, orthrogroups -> orthogroups (Fig 1 caption)

We look forward to receiving your revised manuscript.

Kind regards,

Christopher Toomajian

Academic Editor

PLOS ONE
---

## [Author Response · Author response to Decision Letter 3]

13 Nov 2021

Thank you for catching the last remaining error. I believe I was working of a previous version I had on my computer rather than downloading the file from the editorial center. I believe this was the reason the 3 sentences at L485 were omitted. They have been added and I have reviewed the entire document to ensure their were no other omissions. My apologies.

---

## [Editor Report · Decision Letter 4]

21 Jan 2022

A large accessory genome and high recombination rates may influence global distribution and broad host range of the fungal plant pathogen Claviceps purpurea

PONE-D-21-02186R4

Dear Dr. Broders,

We’re pleased to inform you that your manuscript has been judged scientifically suitable for publication and will be formally accepted for publication once it meets all outstanding technical requirements.

Kind regards,

Christopher Toomajian

Academic Editor

PLOS ONE
---

## [Editor Report · Acceptance letter]

28 Jan 2022

PONE-D-21-02186R4 

A large accessory genome and high recombination rates may influence global distribution and broad host range of the fungal plant pathogen *Claviceps purpurea*

Dear Dr. Broders:

I'm pleased to inform you that your manuscript has been deemed suitable for publication in PLOS ONE. Congratulations! Your manuscript is now with our production department. 

Kind regards, 

on behalf of

Dr. Christopher Toomajian 

Academic Editor

PLOS ONE